# REWARD LEARNING WITH TREES:
# METHODS AND EVALUATION

## ABSTRACT

Recent efforts to learn reward functions from human feedback have tended to use deep neural networks, whose lack of transparency hampers our ability to explain agent behaviour or verify alignment. We explore the merits of learning intrinsically interpretable tree models instead. We develop a recently proposed method for learning reward trees from preference labels, and show it to be broadly competitive with neural networks on challenging high-dimensional tasks, with good robustness to limited or corrupted data. Having found that reward tree learning *can* be done effectively in complex settings, we then consider why it *should* be used, demonstrating that the interpretable reward structure gives significant scope for traceability, verification and explanation.

## 1 INTRODUCTION

For a reinforcement learning (RL) agent to reliably achieve a goal or desired behaviour, this objective must be encoded as a reward function. However, manual reward design is widely understood to be challenging, with risks of under-, over-, and mis-specification leading to undesirable, unsafe and variable outcomes (Pan et al., 2022). For this reason, there has been growing interest in enabling RL agents to learn reward functions from normative feedback provided by humans (Leike et al., 2018). These efforts have proven successful from a technical perspective, but an oft-unquestioned aspect of the approach creates a roadblock to practical applications: reward learning typically uses black-box neural networks (NNs), which resist human scrutiny and interpretation.

For advocates of explainable AI (XAI), this is a problematic state of affairs. The XAI community is vocal about the safety and accountability risks of opaque learning algorithms (Rudin, 2019), but an inability to interpret even the objective that an agent is optimising places us in yet murkier epistemic territory, in which an understanding of the causal origins of learnt behaviour, and their alignment with human preferences, becomes virtually unattainable. Black-box reward learning could also be seen as a missed scientific opportunity. A learnt reward function is a tantalising object of study from an XAI perspective, due to its triple status as (1) an *explanatory* model of revealed human preferences, (2) a *normative* model of agent behaviour, and (3) a *causal* link between the two.

The approach proposed by Bewley & Lecue (2022) provides a promising way forward. Here, human preference labels over pairs of agent behaviours are used to learn tree-structured reward functions (*reward trees*), which are hierarchies of local rules that admit visual and textual representation and can be leveraged to monitor and debug agent learning. In this paper, we adapt and extend the method (including by integrating it with model-based RL agents), and compare it to NN-based reward learning in a challenging aircraft handling domain. We find it to be broadly competitive on both quantitative metrics and qualitative assessments, with our new modification to tree growth yielding significant improvements. The resultant trees are small enough to be globally interpretable ($\approx 20$ leaves), and we demonstrate how they can be analysed, verified, and used to generate explanations.

**The primary contribution of this paper is positive empirical evidence that reward learning can be done effectively using interpretable models such as trees, even in complex, high-dimensional continuous environments.** We also make secondary methodological contributions: improvements to the originally-proposed learning algorithm, as well as metrics and methods for reward evaluation and interpretability that may be useful to others working in what remains a somewhat preparadigmatic field. After reviewing the necessary background and related work in Sections 2 and 3, we present our refinement of reward tree learning in Section 4, and describe how we deploy it online with a model-based agent in Section 5. Section 6 contains our experiments and results, which consider both quantitative and qualitative aspects of learning performance, and an illustrative analysis of learnt tree structures. Finally, Section 7 concludes and discusses avenues for future work.

## 2 BACKGROUND AND RELATED WORK

**Markov Decision Processes (MDPs)**   In this formulation of sequential decision making, the state of a system at time $t$, $s_t \in \mathcal{S}$, and the action of an agent, $a_t \in \mathcal{A}$, condition the successor state $s_{t+1}$ according to dynamics $D : \mathcal{S} \times \mathcal{A} \to \Delta(\mathcal{S})$ ($\Delta(\cdot)$ denotes the set of all probability distributions over a set). A reward function $R : \mathcal{S} \times \mathcal{A} \times \mathcal{S} \to \mathbb{R}$ then outputs a scalar reward $r_{t+1}$ given $s_t$, $a_t$ and $s_{t+1}$. RL uses exploratory data collection to learn action-selection policies $\pi : \mathcal{S} \to \Delta(\mathcal{A})$, with the goal of maximising the expected discounted sum of future reward, $\mathbb{E}_{D,\pi} \sum_{h=0}^{\infty} \gamma^h r_{t+h+1}, \gamma \in [0, 1]$.

**Reward Learning**   In the usual MDP framing, $R$ is an immutable property of the environment, which belies the practical fact that AI objectives originate in the uncertain goals and preferences of fallible humans (Russell, 2019). Reward learning (or modelling) (Leike et al., 2018) replaces hand-specified reward functions with models learnt from humans via revealed preference cues such as demonstrations (Ng et al., 2000), scalar evaluations (Knox & Stone, 2008), approval labels (Griffith et al., 2013), corrections (Bajcsy et al., 2017), and rankings (Christiano et al., 2017). The default use of NNs for reward learning severely limits interpretability; reward trees provide a possible solution.

**XAI for RL (XRL)**   Surveys of XAI for RL (Puiutta & Veith, 2020; Heuillet et al., 2021) divide between intrinsic approaches, which imbue agents with structure such as object-oriented representations (Zhu et al., 2018) or symbolic policy primitives (Verma et al., 2018), and post hoc analyses of learnt representations (Zahavy et al., 2016), including computing feature importance/saliency (Huber et al., 2019). Spatiotemporal scope varies from the local explanation of single actions (van der Waa et al., 2018) to the summary of entire policies via representative trajectories (Amir & Amir, 2018) or critical states (Huang et al., 2018). While most post hoc methods focus on fixed policies, some provide insight into the dynamics of agent learning (Dao et al., 2018; Bewley et al., 2022).

**Explainable Reward Functions**   At the intersection of reward learning and XRL lie efforts to improve human understanding of reward functions and their effects on action selection. While this area is *"less developed"* than other XRL sub-fields (Glanois et al., 2021), a distinction has again emerged between intrinsic approaches which create rewards that decompose into semantic components (Juozapaitis et al., 2019) or optimise for sparsity (Devidze et al., 2021), and post hoc approaches which apply feature importance analysis (Russell & Santos, 2019), counterfactual probing (Michaud et al., 2020), or simplifying transformations (Jenner & Gleave, 2022). Sanneman & Shah (2022) use human-oriented metrics to compare the efficacy of reward explanation techniques. In this taxonomy, reward tree learning is an intrinsic approach, as the rule structure is inherently readable.

**Trees in RL**   Tree models have a long history in RL (Chapman & Kaelbling, 1991; Džeroski et al., 1998; Pyeatt, 2003). Their use is increasingly given an XRL motivation. Applications again divide into intrinsic methods, where an agent's policy (Silva et al., 2020), value function (Liu et al., 2018; Roth et al., 2019) or dynamics model (Jiang et al., 2019) is a tree, and post hoc tree approximations of an existing agent's policy (Bastani et al., 2018; Coppens et al., 2019) or transition statistics (Bewley et al., 2022). Related to our focus on human-centric learning, Cobo et al. (2012) learn tree-structured MDP abstractions from demonstrations and Tambwekar et al. (2021) distil a differentiable tree policy from natural language. While Sheikh et al. (2022) use tree evolution to learn dense intrinsic rewards from sparse environment ones, Bewley & Lecue (2022) are the first to learn and use reward trees in the absence of any ground-truth reward signal, and the first to do so from human feedback.

## 3 PREFERENCE-BASED REWARD LEARNING

We adopt the preference-based approach to reward learning, in which a human is presented with pairs of agent trajectories (sequences of state, action, next state transitions) and expresses which of each pair they prefer as a solution to a given task of interest. A reward function is then learnt to explain the pattern of preferences. This approach is popular in the existing literature (Wirth et al., 2016; Christiano et al., 2017; Lee et al., 2021b) and has a firm psychological basis. Experimental results indicate that humans find it cognitively easier to make relative (*vs.* absolute) quality judgements (Kendall, 1975; Wilde et al., 2020) and exhibit lower variance when doing so (Guo et al., 2018). This is due in part to the lack of requirement for an absolute scale to be maintained in working memory, which is liable to induce bias as it shifts over time (Eric et al., 2007).

We formalise a trajectory $\xi^i$ as a sequence $(\mathbf{x}_1^i, ..., \mathbf{x}_{T^i}^i)$, where $\mathbf{x}_t^i = \phi(s_{t-1}^i, a_{t-1}^i, s_t^i) \in \mathbb{R}^F$ represents a single transition as an $F$-dimensional feature vector. Given $N$ trajectories, $\Xi = \{\xi^i\}_{i=1}^N$, the human provides $K \leq N(N-1)/2$ pairwise preference labels, $\mathcal{L} = \{(i, j)\}_{k=1}^K$, each of which indicates that the $j$th trajectory is preferred to the $i$th (denoted by $\xi^j \succ \xi^i$). Figure 1 (left) shows how a preference dataset $\mathcal{D} = (\Xi, \mathcal{L})$ can be viewed as a directed graph.

Figure 1: Left: The input to preference-based reward learning is a directed graph over a trajectory set $\Xi = \{\xi^i\}_{i=1}^N$, where each edge $(i,j)$ represents a preference $\xi^j \succ \xi^i$. Each member of $\Xi$ is a sequence of points in $\mathbb{R}^F$ (blue connectors show mapping). Right: Application of the four model induction stages from Sections 4.1-4.4 to this example. See Appendix A.3 for an annotated version.

To learn a reward function from $\mathcal{D}$, we must assume a generative model for the preference labels. Typically, it is assumed that the human produces labels in Boltzmann-rational accordance with the sum of rewards (or *return*) output by a latent reward function over the feature space, $R : \mathbb{R}^F \to \mathbb{R}$. This is formalised by adapting the classic preference model of Bradley & Terry (1952):

$$P(\xi^j \succ \xi^i | R) = \frac{1}{1 + \exp(\frac{1}{\beta}(G(\xi^i|R) - G(\xi^j|R)))}, \quad \text{where} \quad G(\xi^i|R) = \sum_{t=1}^{T^i} R(\mathbf{x}_t^i), \quad (1)$$

and $\beta > 0$ is a temperature coefficient. The objective of reward learning is to approximate $R$ within some learnable function class $\mathcal{R}$. This is often formalised as minimising the negative log-likelihood (NLL) loss over $\mathcal{L}$. Wirth et al. (2016) also use the discrete 0-1 loss, which considers only the directions of predicted preferences rather than their strengths. These two losses are defined as:

$$\ell_{\text{NLL}}(\mathcal{D}, R) = \sum_{(i,j) \in \mathcal{L}} -\log P(\xi^j \succ \xi^i | R); \quad \ell_{\text{0-1}}(\mathcal{D}, R) = \sum_{(i,j) \in \mathcal{L}} \mathbb{I}[P(\xi^j \succ \xi^i | R) \le 0.5]. \quad (2)$$

## 4 REWARD TREE INDUCTION

In prior work, $\mathcal{R}$ is the class of linear models $R(\mathbf{x}) = \mathbf{w}^\top \mathbf{x}$ (Sadigh et al., 2017), which have limited expressivity, or deep NNs (Christiano et al., 2017), which resist human interpretation. As an intermediate option, Bewley & Lecue (2022) (BL) propose the reward tree model. Here, the parameter space consists of node-level splitting rules and reward predictions for an axis-aligned decision tree, whose leaves induce a hyperrectangular partition of $\mathbb{R}^F$. While differentiable trees exist (Suárez & Lutsko, 1999), these are of the oblique (c.f. axis-aligned) kind, whose multi-feature rules are far harder to interpret in high dimensions. Therefore, instead of optimising the losses in Equation 2 end-to-end, we use a multi-stage induction method with a proxy objective at each stage. The four stages outlined below, and depicted in Figure 1 (right), depart from BL's original method in several respects. A list of changes, and their performance implications, is given in Appendix A.1.

### 4.1 TRAJECTORY-LEVEL RETURN ESTIMATION

This stage considers the $N$ trajectories as atomic units, and uses the preference graph to construct a vector of return estimates $\mathbf{g} \in \mathbb{R}^N$, which should be higher for more preferred trajectories (blue in Figure 1 (4.1), c.f. red). This is a vanilla preference-based ranking problem, and admits a standard solution. BL use a least squares matrix method to solve for $\mathbf{g}$ under Thurstone's Case V preference model (Gulliksen, 1956). For consistency with prior work, and to avoid an awkward clipping step which biases preference probabilities to enable matrix inversion, we instead use a gradient method to minimise the NLL loss under the Bradley-Terry model. Concretely, the objective for this stage is

$$\underset{\mathbf{g} \in \mathbb{R}^N}{\arg\min} \left[ \sum_{(i,j) \in \mathcal{L}} -\log \frac{1}{1 + \exp(\mathbf{g}^i - \mathbf{g}^j)} \right], \quad \text{subject to} \quad \left\{ \begin{array}{l} \min(\mathbf{g}) = 0 \\ \text{or } \max(\mathbf{g}) = 0 \end{array} \right. \text{and std}(\mathbf{g}) = \beta, \quad (3)$$

where $\beta$ is the mean trajectory length in $\Xi$, $\sum_{i=1}^N T^i / N$. The min-or-max constraint ensures that all return estimates have the same sign (positive or negative), which aids both policy learning and interpretability (see Appendix A.2). We first optimise the NLL loss by gradient descent with the Adam optimiser (Kingma & Ba, 2014), then apply shift and scale factors to meet the two constraints.

### 4.2 LEAF-LEVEL REWARD PREDICTION

The vector $\mathbf{g}$ estimates trajectory-level returns, but the aim of reward learning is to decompose these into sums of rewards for the constituent transitions, then generalise this to make reward predictions for unseen data (e.g. novel trajectories executed by a learning agent). BL's contribution is to do this using a tree model $\mathcal{T}$, consisting of a hierarchy of rules that partition the transition-level feature space $\mathbb{R}^F$ into $L_{\mathcal{T}}$ hyperrectangular subsets called *leaves*. Each leaf $l \in \{1..L_{\mathcal{T}}\}$ is associated with

a reward prediction $\mathbf{r}_l$ as follows. Let the function $\mathrm{leaf}_{\mathcal{T}} : \mathbb{R}^F \to \{1..L_{\mathcal{T}}\}$ map a feature vector $\mathbf{x} \in \mathbb{R}^F$ to the leaf in which it resides by propagating it through the rule hierarchy. $\mathbf{r}_l$ is defined as an average over $\mathbf{g}$, weighted by the proportion of timesteps that each trajectory in $\Xi$ spends in $l$:

$$\mathbf{r}_l = \sum_{i=1}^N \frac{\mathbf{g}^i}{T^i} \frac{\sum_{t=1}^{T^i} \mathbb{I}[\mathrm{leaf}_{\mathcal{T}}(\mathbf{x}_t^i) = l]}{\sum_{j=1}^N \sum_{t=1}^{T^j} \mathbb{I}[\mathrm{leaf}_{\mathcal{T}}(\mathbf{x}_t^j) = l]}. \tag{4}$$

The effect of Equation 4 is to assign higher reward to leaves that contain more timesteps from trajectories with high $\mathbf{g}$ values. Predicting the reward for an arbitrary unseen feature vector $\mathbf{x} \in \mathbb{R}^F$ then involves simply looking up the reward of the leaf in which it resides: $R_{\mathcal{T}}(\mathbf{x}) = \mathbf{r}_{\mathrm{leaf}_{\mathcal{T}}(\mathbf{x})}$.

While ostensibly naïve, BL find that this time-weighted credit assignment is more robust than several more sophisticated alternatives. It reduces the number of free parameters in subsequent induction stages, permits fast implementation, and provides an intuitive interpretation of predicted reward that is traceable back to a $\mathbf{g}$ value and timestep count for each $\xi^i \in \Xi$. Figure 1 (4.2) shows how 4, 2 and 3 timesteps from $\xi^1$, $\xi^3$ and $\xi^4$ are averaged over to yield the reward prediction for one leaf (indicated by the orange shading). For more intuition on this stage, see the annotated figure in Appendix A.3.

### 4.3    TREE GROWTH

Recall that the objective of preference-based reward learning is to find a reward model that optimises a measure of fidelity to $\mathcal{D}$, such as the losses in Equation 2. When the model is a tree, this is achieved by the discrete operations of growth (adding partitioning rules) and pruning (removing rules). Given a tree $\mathcal{T}$, a new rule has the effect of splitting the $l$th leaf with a hyperplane at a location $c \in \mathcal{C}_f$ along the $f$th feature dimension (where $\mathcal{C}_f \subset \mathbb{R}$ is a set of candidate split thresholds, e.g. all midpoints between unique values in $\Xi$). Let $\mathcal{T} + [lfc]$ denote the newly-enlarged tree. Splitting recursively creates an increasingly fine partition of $\mathbb{R}^F$. Figure 1 (4.3) shows an example with 23 leaves.

A central issue is the criterion for selecting the next rule to add. BL use the proxy objective of minimising the local variance of reward predictions, which is exactly the CART algorithm (Breiman et al., 2017). While very fast, this criterion is only loosely aligned with fidelity to the preferences in $\mathcal{D}$. We propose the more direct criterion of greedily maximising the immediate reduction in $\ell_{0\text{-}1}$:

$$\mathrm{argmax}_{1 \leq l \leq L_{\mathcal{T}}, \, 1 \leq f \leq F, \, c \in \mathcal{C}_f} \left[ \ell_{0\text{-}1}(\mathcal{D}, R_{\mathcal{T}}) - \ell_{0\text{-}1}(\mathcal{D}, R_{\mathcal{T}+[lfc]}) \right]. \tag{5}$$

In Section 6, we show that switching to this bespoke criterion consistently improves performance. Its implementation involves a major reformulation of the tree growth algorithm; we provide vectorised, just-in-time compiled code for this in the Supplementary Material. Recursive splitting stops when no reduction in $\ell_{0\text{-}1}$ can be achieved by any single split, or a tree size limit $L_{\mathcal{T}} = L_{\max}$ is reached.

### 4.4    TREE PRUNING

Growth is followed by a pruning sweep which reduces the size of the tree by rule removal. Such reduction is beneficial for both performance (Tien et al. (2022) find that limiting model capacity lowers the risk of causal confusion in preference-based reward learning) and human comprehension (in the language of Jenner & Gleave (2022), it is a form of *"processing for interpretability"*). Given a tree $\mathcal{T}$, one pruning operation has the effect of merging two leaves into one by removing the rule at the common parent node. Let $\mathbb{T}$ denote the sequence of nested subtrees induced by pruning the tree recursively back to its root, at each step removing the rule that minimises the next subtree's $\ell_{0\text{-}1}$. We select the $\mathcal{T} \in \mathbb{T}$ that minimises $\ell_{0\text{-}1}$, additionally regularised by a term that encourages small trees: $\mathrm{argmin}_{\mathcal{T} \in \mathbb{T}}[\ell_{0\text{-}1}(\mathcal{D}, R_{\mathcal{T}}) + \alpha L_{\mathcal{T}}]$, where $\alpha \geq 0$. Note that even with $\alpha = 0$ pruning may still yield a reduced tree, as unlike in traditional decision tree induction, the effect of individual rules on $\ell_{0\text{-}1}$ depends on the order in which they are added or removed. In the example in Figure 1 (4.4), pruning yields a final tree with 3 leaves, for which illustrative leaf-level reward predictions are shown.

## 5    ONLINE LEARNING SETUP

### 5.1    ITERATED POLICY AND REWARD LEARNING

Sections 3 and 4 do not discuss the origins of the trajectories $\Xi$, or how reward learning relates to the downstream objective of learning a policy for the underlying task. Following most recent work since Christiano et al. (2017), we resolve both questions with an online bootstrapped approach. Assuming an episodic MDP, the $i$th episode of policy learning produces a new trajectory $\xi^i$ to add to $\Xi$. We immediately connect $\xi^i$ to the preference graph by asking the human to compare it to $K_{\mathrm{batch}}$ random trajectories from the existing set (while Sadigh et al. (2017) and others have proposed active querying schemes, that is not our focus here, and this simple strategy performs satisfactorily). We then update the reward tree on the full preference graph via the four stages given in Section 4. We

find that BL's original method of starting growth from the current state of the tree causes lock-in to poor initial solutions, so instead re-grow from scratch on each update. The rule structure nonetheless tends to stabilise, as the enlarging preference graph becomes increasingly similar for later updates. For the $(i + 1)$th episode, the policy learning agent then attempts to optimise for the newly-updated reward. By iterating this process up to a total preference budget $K_{\max}$ and/or episode budget $N_{\max}$, we hope to converge to both a reward tree that reflects the human's preferences, and an agent policy that satisfies those preferences. Appendix A.4 contains pseudocode for the online algorithm.

## 5.2 INTEGRATION WITH MODEL-BASED RL

Reward learning methods are generally agnostic to the structure of the policy learning agent; this modularity is hailed as an advantage over other human-agent teaching paradigms (Leike et al., 2018). In line with most recent works, BL use a model-free RL agent, specifically soft actor-critic (SAC) (Haarnoja et al., 2018). However, other works (Reddy et al., 2020; Rahtz et al., 2022) use model-based RL (MBRL) agents that leverage learnt dynamics models and planning. MBRL is attractive in the reward learning context because it disentangles the predictive and normative aspects of decision-making. Since (assuming no changes to the environment) dynamics remain stationary during online reward learning, the amount of re-learning required is reduced and along with it, the risk of pitfalls such as manipulation (Armstrong et al., 2020) and premature convergence. Additionally, MBRL can be very data-efficient; we find that switching from SAC to a model-based algorithm called PETS (Chua et al., 2018) reduces environment interaction during reward learning by orders of magnitude, and cuts wall-clock runtime (see Appendix B). PETS selects actions by decision-time planning through a learnt dynamics model $D' : \mathcal{S} \times \mathcal{A} \to \Delta(\mathcal{S})$ up to a horizon $H$. In state $s$, planning searches for a sequence of $H$ future actions that maximise return under the current reward model:

$$\underset{(a_0,\dots,a_{H-1})\in\mathcal{A}^H}{\operatorname{argmax}} \mathbb{E}_{D'}\left[\sum_{h=0}^{H-1}\gamma^h R_{\mathcal{T}}(\phi(s_h, a_h, s_{h+1}))\right], \text{ where } s_0 = s, \ s_{h+1} \sim D'(s_h, a_h). \quad (6)$$

The first action $a = a_0$ is executed, and then the agent re-plans on the next timestep. In practice, $D'$ is an ensemble of probabilistic NNs, the expectation over $D'$ is replaced by a Monte Carlo estimate, and the optimisation is approximated by the iterative cross-entropy method.

## 6 EXPERIMENTS AND RESULTS

In this section, we combine quantitative and qualitative evaluations to assess the performance of reward tree learning, specifically in comparison to the standard approach of using NNs. We also illustrate how the intrinsic interpretability of reward trees allows us to analyse what they have learnt. Our experiments focus on an aircraft handling domain (Figure 2), in which an agent must manoeuvre an aircraft (the *ego jet*, EJ) in a desired manner relative to a second *reference jet* (RJ) whose motion, if any, is part of the environment dynamics. We consider three tasks: **Follow** (turn to fly in formation with RJ on a linear path); **Chase** (maintain distance/line of sight to RJ as it turns randomly); and **Land** (approach a runway using RJ as a reference). These tasks are useful test cases for reward learning, as each has a large space of plausible reward functions, which may reflect the divergent priorities and stylistic preferences of aeronautical experts. Such expert knowledge is often tacit and difficult to codify (Sternberg & Horvath, 1999), motivating a learning-based approach. Appendix C contains a broader justification of this experimental domain alongside implementation details.

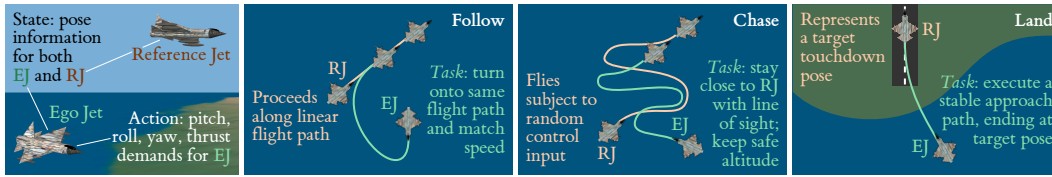

Figure 2: State-action space of aircraft handling domain, and diagrams of Follow/Chase/Land tasks.

In place of costly human-in-the-loop evaluation, our experiments use synthetic *oracle* preferences with respect to nominal reward functions of varying complexity, which are given in Appendix C.3. This approach is popular (Griffith et al., 2013; Christiano et al., 2017; Reddy et al., 2020; Lindner et al., 2021) as it enables scalable systematic comparison, with the ability to quantify performance (and in our case, appraise learnt trees) in terms of reconstruction of a known ground truth. However, emulating a human with an oracle that responds with perfect rationality is unrealistic (Lee et al., 2021a). For this reason, Section 6.3 examines the performance impacts of noisy and myopic oracles, and a restricted data budget. Experimental details and hyperparameters are given in Appendix D.

### 6.1 QUANTITATIVE PERFORMANCE

We evaluate online reward learning with PETS using trees with the $\ell_{0\text{-}1}$ split criterion, baselined against BL's original variance criterion, as well as the de facto standard of NN reward learning (see Appendix D.4 for details). We use $K_{\max} = 1000$ preferences over $N_{\max} = 200$ online trajectories, and run 10 repeats. As a headline statistic, we report the *oracle regret ratio* (ORR): the median drop in oracle return of PETS agents deployed using each trained reward model compared with directly using the oracle reward, as a fraction of the drop to a random policy (lower is better). Below are the median (top) and minimum (bottom) ORR values across the 10 repeats for each task-model pairing:

| | Follow | | | Chase | | | Land | |
|---|---|---|---|---|---|---|---|---|
| NN | Tree(0-1) | Tree (var) | NN | Tree (0-1) | Tree (var) | NN | Tree (0-1) | Tree (var) |
| .000 | .120 | .284 | $-.030$ | .040 | .126 | .014 | .050 | .062 |
| $-.010$ | .057 | .158 | $-.051$ | $-.011$ | .065 | $-.030$ | .011 | .010 |

We observe that: 1) NN reward learning is strong on all tasks; 2) switching to a reward tree induces a small but variable performance hit; 3) $\ell_{0\text{-}1}$ splitting outperforms the variance-based method; and 4) both NN and tree models sometimes exceed the direct use of the oracle (negative ORR). This has been observed before (Cao et al., 2021) and may be due to improved shaping in the learnt reward.

Figure 3 expands these results with more metrics, revealing subtler trends not captured by headline ORR values. Metrics are plotted as time series over the 200 learning episodes (sliding-window medians and interquartile ranges across repeats). In the left column (**a**), the ORR of online trajectories shows how agent performance converges. **For Follow, there is a gap between the models**, with $\ell_{0\text{-}1}$ splitting clearly aiding performance but still lagging behind the NNs. **The learning curves for Chase and Land are more homogeneous**, and the NNs reach only slightly lower asymptotes. For the reward tree models, (**b**) shows how the number of leaves changes over time. The variance-based trees tend to grow rapidly initially before stabilising or shrinking, while **the $\ell_{0\text{-}1}$ trees enlarge more conservatively, suggesting this method is less liable to overfit** to small preference graphs. Trees of a readily-interpretable size ($\approx 20$ leaves) are produced for all tasks; it is possible that performance could be improved by independently tuning the size regulariser $\alpha$ per task. (**c**) shows $\ell_{0\text{-}1}$ over time, which tends to increase as the growing preference graph presents a harder reconstruction problem, though the shape of all curves suggests convergence (note that random prediction gives $\ell_{0\text{-}1} = 0.5$). For Follow and Land, **the trees that directly split on $\ell_{0\text{-}1}$ actually perform better than the NNs; they more accurately predict the direction of preferences in the graph**. The fact that this does not translate into lower ORR indicates that the problems of learning a good policy and replicating the preference dataset are not identical, a point made by Lindner et al. (2021). In the final two columns, we follow Gleave et al. (2021) in performing an unbiased, *policy-invariant* comparison of the models by correlating their outputs with the oracle reward functions on common evaluation datasets (see Appendix D.5 for dataset creation). We compute online correlations with the oracles in terms of both transition-level rewards (**d**) and the ordinal ranking of trajectories by return (**e**), the latter via the Kendall (1938) $\tau$ coefficient. The curves subtly differ, indicating that it is possible to reconstruct trajectory rankings (and by extension, any pairwise preferences) to a given accuracy with varying fidelity at the individual reward level. However, the common overall trend is that **$\ell_{0\text{-}1}$-based trees outperform variance-based ones, with NNs sometimes improving again by a smaller margin, and sometimes bringing no added benefit**. Moving top-to-bottom down the tasks, the gap between models reduces from both sides; NN performance worsens while variance-based trees improve.

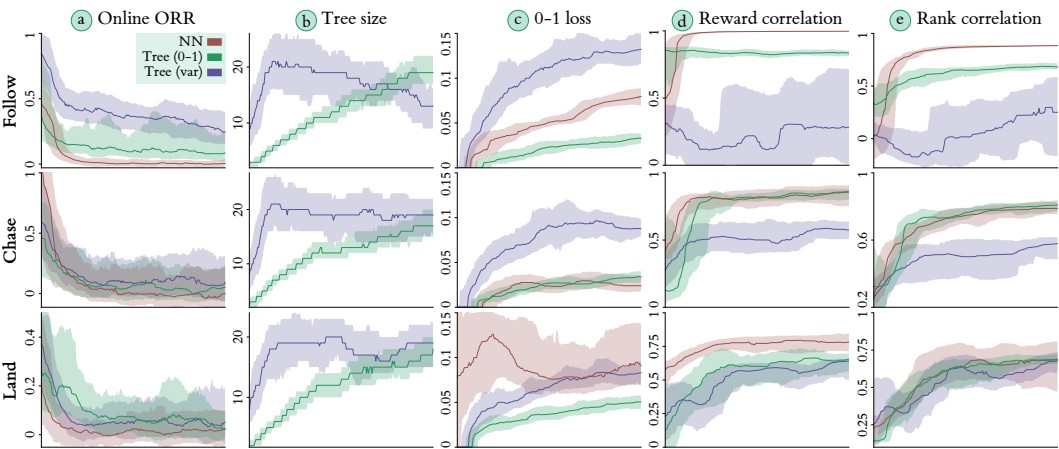

Figure 3: Time series of metrics for online NN- and tree-based reward learning on all three tasks.

A potentially important factor in these experiments is that the oracle reward for Follow is a simple linear function, while the other two contain progressively more terms and discontinuities (see Appendix C.3). A trend suggested by these results is thus that **the performance gap between NNs and reward trees** (on both ORR and correlation metrics) **reduces as the ground truth reward becomes more complex and nonlinear**. Further experiments would be needed to test this hypothesis.

## 6.2 VISUAL TRAJECTORY INSPECTION

While useful for benchmarking, quantitative metrics provide little insight into the structure of the learnt solutions. They would also mostly be undefined when learning from humans since the ground truth reward is unknown. We therefore complement them with a visual analysis of induced agent behaviour. Figure 4 plots 500 trajectories of PETS agents using the best repeat by ORR for each task-model combination, across a range of features as well as time (see Appendix C.2 for feature definitions). Dashed curves show the trajectory with the highest predicted return according to each model. We also show trajectories for PETS agents with direct oracle access, and for random policies.

The high-level trend is that all models are far closer to the oracle than random, with few examples of obviously incorrect behaviour (highlighted in red, due to colouring by ORR). **While the NNs induce trajectories that are almost indistinguishable from the oracle, the $\ell_{0\text{-}1}$-based reward trees lag not far behind. The variance-based trees produce more anomalies.** Successes of the $\ell_{0\text{-}1}$ trees include the execution of Follow with a single banked turn before straightening up, as shown by the up error time series (**a**). Indeed, the trajectories for this model are almost imperceptibly different from those of the NN, a result which is belied by the mediocre ORR of $0.158$. This underlines the importance of joint quantitative-qualitative evaluation. For Chase (**b**), the $\ell_{0\text{-}1}$ tree has learnt to keep the agent narrowly above the altitude threshold $\text{alt} < 50$, below which the oracle reward is strongly negative (see Appendix C.3). The threshold is violated in only eight of $500$ trajectories ($1.6\%$). For Land, the $\ell_{0\text{-}1}$ tree replicates the oracle in producing a gradual reduction in $\text{alt}$ (**c**) while usually keeping $\text{pitch}$ close to $0$ (**d**), although the distribution of $\text{roll}$ values is less narrow.

In contrast, the agent using the variance-based tree for Follow sometimes fails to reach the target position (**e**; red trajectories), and also does not reliably straighten up to reduce up error (**f**). For Chase, the altitude threshold does not appear to have been learnt precisely, and lower-altitude trajectories often fail to close the distance to RJ (**g** and **h**; red trajectories). For Land, the variance-based tree gives a later and less smooth descent (**i**), and less consistent pitch control (**j**), than the NN or $\ell_{0\text{-}1}$-based tree, although all models produce a somewhat higher altitude profile than the oracle.

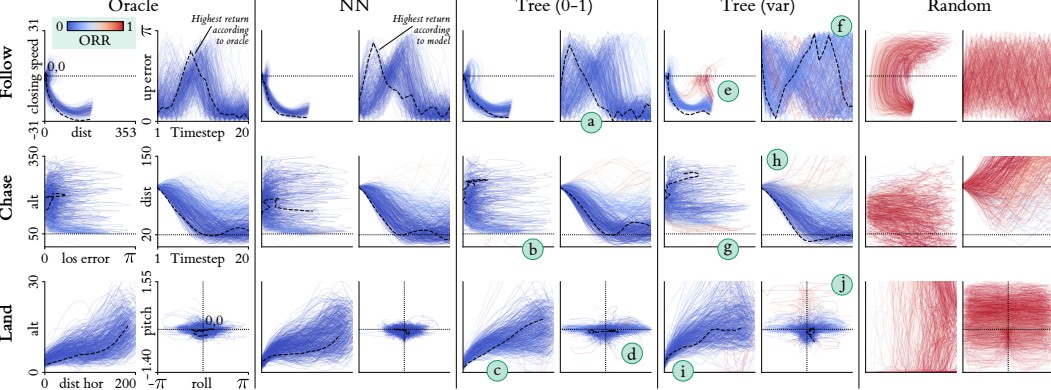

Figure 4: Agent trajectories using the best models by ORR, with oracle and random for comparison.

## 6.3 SENSITIVITY ANALYSIS

It is important to consider how learning performance degrades with reduced or corrupted data. In Figure 5, we evaluate the effect of varying the number of preferences $K_{\max}$ (with fixed $N_{\max} = 200$) and trajectories $N_{\max}$ (with fixed $K_{\max} = 1000$) on reward learning with NNs and $\ell_{0\text{-}1}$-splitting trees. Following Lee et al. (2021a), we also create more human-like preference data via two modes of oracle *irrationality*: preference noise (by using a nonzero Boltzmann temperature $\beta$ to give a desired error rate on the coverage datasets) and a myopic recency bias (by exponentially discounting earlier timesteps when evaluating trajectory returns). We run five repeats for all cases, and report the medians and interquartile ranges of ORR (lower is better) and rank correlation (higher is better).

Both NN and tree models exhibit good robustness with respect to all four parameters. Although NNs remain superior in most cases, the gap varies, and is often reduced compared to the base

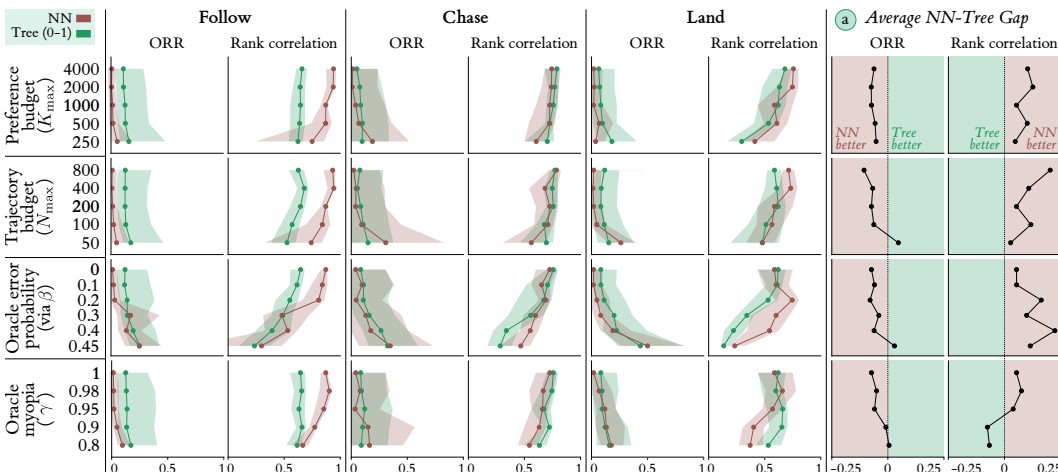

Figure 5: Comparative sensitivity analysis of reward learning with NNs and trees.

cases (bold labels). The budget sensitivity is low, with little improvement for $K_{\max} > 1000$ and $N_{\max} > 200$, and no major drop even with $25\%$ of the data as the base case. For all tasks, the oracle error probability can increase to around $20\%$ before significant drops in performance are observed. This is a promising indicator of the transferability of reward tree learning to imperfect human data. Another general observation is that the trends for trees are somewhat smoother than for NNs, with fewer sharp jumps and fewer instances of very high spread across the five repeats.

In the right column (**a**), we summarise these results by taking the difference between the NN and tree metrics, and averaging across the three tasks. In all cases aside from rank correlation with $\beta > 0$, the NN-tree gap tends to become more favourable to the tree models as the varied parameter becomes more challenging (top-to-bottom). This sensitivity analysis thus indicates that **reward trees are at least as robust to difficult learning scenarios as NNs, and may even be slightly more so**.

## 6.4 TREE STRUCTURE ANALYSIS

Thus far we have shown that reward learning with $\ell_{0\text{-}1}$-based trees can be competitive with NNs, but not quite as performant overall. We now turn to a concrete advantage which may tip practical tradeoffs in its favour: the ability to interpret the learnt model, and analyse how its structure arises from the underlying preference graph. In this section we favour depth over breadth, so focus on the single best tree by ORR on the Chase task. The analysis in Figure 6 is divided into sections (**a** – **d**):

(**a**) This reward tree has 17 leaves. The oracle reward, printed below, uses four features, all of which are used in the tree in ways that are broadly aligned (e.g. lower `los error` leads to leaves with higher reward). The model has learnt the crucial threshold `alt < 50`, correctly assigning low reward when it is crossed. This explains why we observe rare violations of the altitude threshold in Figure 4. However, it has not learnt the ideal distance to RJ, `dist = 20`, with $43.3$ being the lowest value used in a rule. This could be because the underlying preference graph lacks sufficient preferences to make this distinction; adopting an active querying scheme may help to discover such subtleties efficiently. Other features besides those used by the oracle are present in the tree, indicating some causal confusion (Tien et al., 2022). This may not necessarily harm agent performance, as it could provide beneficial shaping (e.g. penalising positive `closing speed`, which indicates increasing distance to RJ). That may indeed be the case for this model since ORR is actually negative.

(**b**) We plot the tree's predicted reward against the oracle reward for all timesteps in the online trajectories (correlation $= 0.903$). The predictions for each leaf lie along a horizontal line. Most leaves, including 1 and 2, are well-aligned on this data because their oracle reward distributions are tightly concentrated around low/high averages respectively (note that the absolute scale is irrelevant here). Leaf 16 has a wider oracle reward distribution, with several negative outliers. An optimal tree would likely split this leaf further, perhaps using the `alt < 50` threshold. The one anomaly is leaf 13, which contains just a single timestep from $\xi^{77}$. This trajectory is the eighth best in the dataset by oracle return, but this leaf assigns that credit to a state that seemingly does not merit it, as the distance to RJ is so high (`dist > 73`). This may be an example of suboptimal reward learning, but the fact that its origin can be pinpointed precisely is a testament to the value of interpretability.

(**c**) We leverage the tree structure to produce a human-readable explanation of reward predictions for a single trajectory, which may be of value to an end user (e.g. a pilot). We consider $\xi^{191}$, a rare

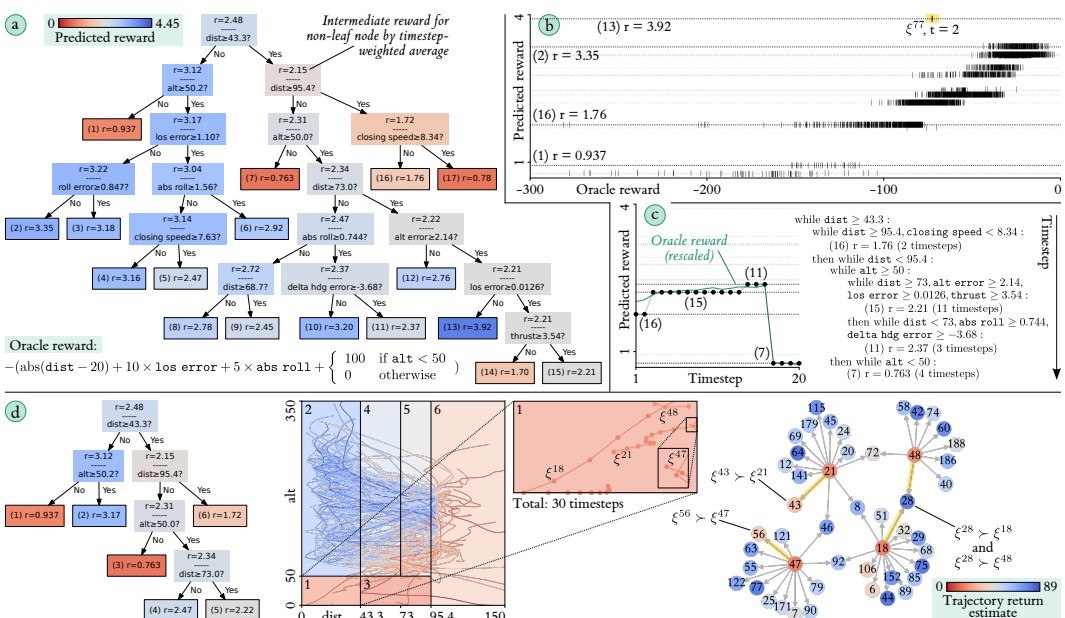

Figure 6: Analysis of a reward tree learnt for the Chase task.

case that violates the altitude threshold. The time series of reward shows that the 20 timesteps are spent in leaves 16, 15, 11 and 7. Rescaled oracle rewards are overlaid in teal, and show that the model's predictions are well-aligned. To the right, we translate this visualisation into a textual form, similar to a nested program. Read top-to-bottom, the text indicates which rules of the tree are active at each timestep, and the effect this has on predicted reward. This trajectory starts fairly positively, with reward gradually increasing over the first 16 timesteps as dist is reduced to between 43.3 and 73, but then falls dramatically when the alt < 50 threshold is crossed. We are unaware of any method that could extract such a compact explanation of sequential predictions from an NN.

(**d**)  We isolate a subtree, starting at the root node, that splits only on dist and alt. We give a spatial representation of the subtree, and how it is populated by the 200 online trajectories, using a 2D partition plot analogous to those in Figure 1. Zooming into leaf 1, which covers cases where the altitude threshold is violated, we see that it contains a total of 30 timesteps across four trajectories. By Equation 4, the low reward for this leaf results from a weighted average of the return estimates for these four trajectories, which in turn (by Equation 3) are derived from the preference graph. We can use this inverse reasoning to ask *why* this leaf has much lower reward than its sibling (leaf 2 of the subtree). A proximal explanation comes by filtering the graph for preferences that specifically compare trajectories that visit those two leaves. 49 such preferences exist, and in all cases, the oracle prefers the trajectory that does not visit leaf 1. Some of these preferences may be more practically salient than others. For example, we might highlight trajectories that feature more than once (e.g. $\xi^{28}$ is preferred to both $\xi^{18}$ and $\xi^{48}$), or cases where trajectories with low overall return estimates are nonetheless preferred to those in leaf 1 (e.g. $\xi^{43} \succ \xi^{21}$ and $\xi^{56} \succ \xi^{47}$). We believe that much more could be done to extend this framework for traceable explanation of preference-based reward.

## 7  CONCLUSION AND FUTURE WORK

Reward learning with trees provides a promising alternative to black-box NNs, and could enable more trustworthy and verifiable agent alignment. Through oracle experiments on high-dimensional tasks, we show that reward trees with around 20 leaves can achieve quantitative and qualitative performance close to that of NNs, with a more direct split criterion bringing consistent improvements. We find evidence that the NN-tree gap reduces as the ground truth reward becomes more nonlinear, and remains stable or reduces further in the presence of limited or corrupted data. While practical applications may accept some loss in performance for a gain in interpretability, further algorithmic improvements should be sought, including to move beyond locally-greedy split criteria. However, our immediate aim is to develop an end-to-end framework for explainable model-based agents with preference-based reward trees (roughly: planning can be reframed as comparing alternative paths through the discrete leaves of the tree). Having established this framework, we then intend to evaluate reward learning and explanation with real human preferences in the aircraft handling domain.

## ETHICS STATEMENT

Reward learning from feedback is a technique for improving the alignment of learning agents with human preferences, by replacing the rigidity of explicit reward design with a dynamic interaction with a human-in-the-loop. As such, its successful use can benefit the performance, reliability and safety of these learning systems, which has the potential to deliver immensely positive ethical value. The contribution of reward trees is to render the process of reward learning more human-interpretable, and thus easier to explain, debug and verify. We believe this can deliver a further reduction in ethical risk through the identification and mitigation of unforeseen consequences.

Preference-based reward tree learning has many diverse applications. The aircraft handling domain used in our evaluation was selected to provide a good balance of technical complexity, task diversity, industrial relevance, intuitiveness for our intended readership, and transferability to other domains such as land and sea transportation. In addition to this wide range of related use cases, as well as civilian uses of aviation itself (e.g. aerobatics), we acknowledge that the ability to interactively learn aircraft control policies may have applications in the defence sector. The three concrete tasks of Follow, Chase and Land are largely application-neutral, with no implication of harm, and are concerned solely with the safe and human-like control of aircraft in environments with other aircraft. Any general learning technique such as ours is, however, fundamentally dual-use, and transparency about this fact seems to us the best mitigation of the ethical risk. It is vital that anyone intending to use or develop our method continues to do so in an ethically responsible manner.

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

# A   METHODOLOGICAL DETAILS

## A.1   LIST OF CHANGES TO BEWLEY & LECUE'S ORIGINAL METHOD

- Change from Thurstone to Bradley-Terry preference model and least squares matrix method to gradient descent on $\ell_{\text{NLL}}$;
    - *Motivation*: Consistency with prior work.
    - *Performance implications*: Thurstone $\rightarrow$ Bradley-Terry minimally affects results. Matrix method $\rightarrow$ gradient descent eliminates a bias caused by preference clipping.
    - *Computational implications*: Thurstone $\rightarrow$ Bradley-Terry improves computational efficiency (bypasses inverse normal CDF computation). Matrix method $\rightarrow$ gradient descent tends to slightly increase runtime, but this depends on preference dataset size.
- Add scale and sign constraints to return estimates;
    - *Motivation / implications*: Scaling results in leaf-level reward predictions having a consistent scale with a standard deviation close to 1. This is intended to aid human readability but has no effect on performance. For sign constraints, see Appendix A.2.
- Change from variance-based to $\ell_{\text{0-1}}$-based split criterion during tree growth;
    - *Motivation*: Hypothesised performance gain; new split criterion is more directly aligned with the objective of preference reconstruction.
    - *Performance implications*: Significantly improves preference construction and quantitative/qualitative agent performance on evaluation tasks; see Section 6.
    - *Computational implications*: More costly as variance computation has an extremely efficient iterative implementation. However, we have developed optimised code for the new splitting method using just-in-time compilation; see Supplementary Material.
- Deploy online with a model-based (PETS) RL agent instead of model-free (SAC);
    - *Motivation / implications*: See Section 5.2 and Appendix B.
- Always use latest online trajectory in all pairs during each preference batch;
    - *Motivation*: Acts as a simple form of active sampling to correct reward overestimation; model-based planning is liable to exploit any behaviours with inappropriately high current reward, which can then be immediately corrected by a negative preference.
    - *Performance implications*: Performs similarly to BL's active sampling method, which up-weights trajectories in $\Xi$ with high predicted return.
    - *Computational implications*: Less expensive than BL's method; no need to recompute return predictions for all trajectories in $\Xi$ on each batch.
        - * Note: New method would be less effective with a model-free agent where policy updates are gradual; can rely less on agent immediately exploiting current reward.
- Regrow tree from scratch on each update.
    - *Motivation*: Prevents rule structure from prematurely converging to local minima.
    - *Performance implications*: Early experiments indicated that premature convergence problem is mitigated by this change, resulting in more sustained improvements in reward fidelity and agent performance.
    - *Computational implications*: Since splits are evaluated and made per update step, computation time is increased. However, when typical post-pruning tree size $L_{\mathcal{T}}$ ($\approx 20$ in our experiments) is small compared with $L_{\text{max}}$ ($= 100$), this increase is fractional, and contributes only a few percentage points to overall runtime.

## A.2 SIGN CONSTRAINT FOR RETURN ESTIMATES

Applying a sign constraint to the trajectory-level return estimates means that rewards output by a reward tree (via Equation 4) are also all either positive or negative. This has no effect on any measure of preference reconstruction since preferences are invariant to affine transformations of an underlying utility function. However, we find it brings two distinct benefits:

- Enabling the prevention of perverse incentives for agents to terminate or elongate episodes in tasks with termination conditions (negative rewards on non-terminal transitions incentivise termination, while positive rewards incentivise elongation).
- Simplifying the manual interpretation of tradeoffs between rewards from different leaves of a tree (understanding the relative impacts of "more of a negative reward" and "less of a positive reward" requires the awkward mental juggling of negatives).

For the task with a termination condition in this paper (Land), we use negative rewards (max = 0 constraint) to disincentivise episode elongation, because termination is generally indicative of success. For the two fixed-length tasks (Follow and Chase) we default to using positive rewards (min = 0 constraint). Although this is arbitrary, our own experience is that positive rewards make for somewhat more intuitive interpretation of the tree structure, and its effect on agent actions. We stress that this is purely anecdotal; the relative human interpretability of positive, negative and mixed-sign rewards would be a worthy subject for deeper empirical investigation.

## A.3 ANNOTATED VERSION OF FIGURE 1

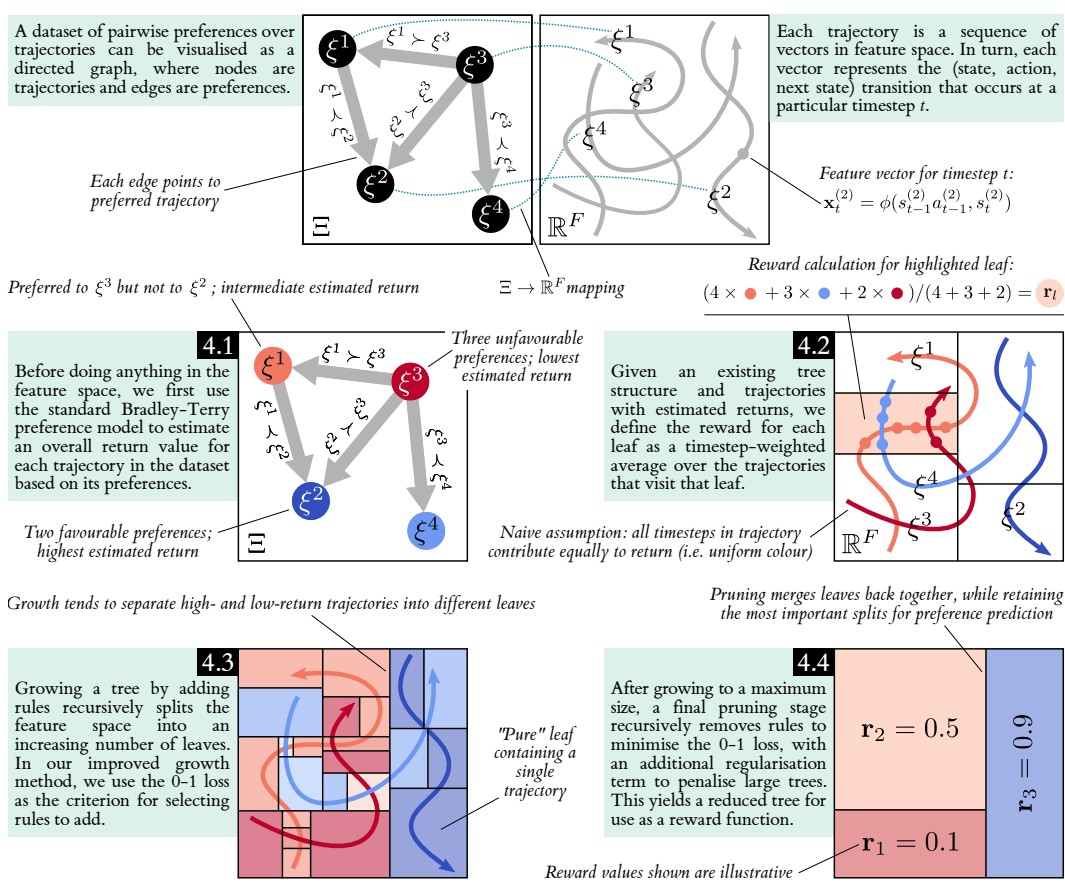

Figure A1: Annotated version of Figure 1.

A.4 PSEUDOCODE FOR ONLINE ALGORITHM

---

**Algorithm 1** Online preference-based reward tree learning

---

    **Inputs:** Possibly pre-trained dynamics model $D'$, feature function $\phi$, trajectory budget $N_{\max}$, preference budget $K_{\max}$, tree size limit $L_{\max}$, tree size regularisation $\alpha$

1: Initialise empty preference graph $\Xi \leftarrow \emptyset$, $\mathcal{L} \leftarrow \emptyset$
2: Initialise one-leaf tree $\mathcal{T}$ with $\mathbf{r} \leftarrow [0]$
3: **for** $i \in \{1, ..., N_{\max}\}$ **do**
4:     Initialise time $t \leftarrow 0$ and environment state $s_0^i$
5:     **while** episode not yet terminated **do**         ▷ Model-based trajectory generation (Sec.5.2)
6:         Compute action $a_t^i$ using PETS algorithm with $D'$ and rewards via Equation 4
7:         Send $a_t^i$ to environment and get next state $s_{t+1}^i$
8:         Update $D'$ on recent transitions         ▷ May not be required; see Appendix D.3
9:         $\mathbf{x}_{t+1}^i \leftarrow \phi(s_t^i, a_t^i, s_{t+1}^i)$
10:         $t \leftarrow t + 1$
11:     **end while**
12:     $\xi^i \leftarrow (\mathbf{x}_1^i, ..., \mathbf{x}_{T^i}^i)$
13:     $K_{\text{batch}} \leftarrow \min((K_{\max} - |\mathcal{L}|)/(N_{\max} + 1 - i), |\Xi|)$
14:     **for** $k \in \{1, ..., K_{\text{batch}}\}$ **do**         ▷ Preference batch collection (Sec 3)
15:         Sample $\xi^j$ from $\Xi$ uniformly without replacement
16:         Query human for preference $\xi^i \succ \xi^j$ or $\xi^j \succ \xi^i$
17:         $\mathcal{L} \leftarrow \mathcal{L} \cup \begin{cases} \{(i,j)\} & \text{if } \xi^j \succ \xi^i \\ \{(j,i)\} & \text{otherwise} \end{cases}$
18:     **end for**
19:     $\Xi \leftarrow \Xi \cup \{\xi^i\}$
20:     **if** $|\mathcal{L}| > 0$ **then**
21:         Compute $\mathbf{g}$ via Equation 3         ▷ Trajectory-level return estimation (Sec 4.1)
22:         Initialise one-leaf tree $\mathcal{T}$
23:         $\mathcal{C} \leftarrow$ midpoints between per-feature unique values in $\Xi$
24:         **while** $L_{\mathcal{T}} < L_{\max}$ **do**         ▷ Tree growth (Section 4.3)
25:             **for** $l \in \{1, ..., L_{\mathcal{T}}\}$ **do**
26:                 **for** $f \in \{1, ..., F\}$ **do**
27:                     **for** $c \in \mathcal{C}_f$ **do**
28:                         Compute $\ell_{0\text{-}1}$ reduction for $\mathcal{T} + [lfc]$ via Equation 5
29:                     **end for**
30:                 **end for**
31:             **end for**
32:             **if** $\max(\ell_{0\text{-}1} \text{ reduction}) \leq 0$ **then**
33:                 **break**         ▷ Stop tree growth early
34:             **end if**
35:             $l, f, c \leftarrow \text{argmax}(\ell_{0\text{-}1} \text{ reduction})$
36:             $\mathcal{T} \leftarrow \mathcal{T} + [lfc]$
37:         **end while**
38:         $\mathbb{T} = ()$
39:         **while** $L_{\mathcal{T}} > 1$ **do**         ▷ Tree pruning (Section 4.4)
40:             **for** $l \in \{1, ..., L_{\mathcal{T}}\}$ **do**
41:                 Compute $\ell_{0\text{-}1}$ reduction for $\mathcal{T} - [l]$         ▷ $\mathcal{T} - [l]$ denotes pruning $l$th leaf
42:             **end for**
43:             $l \leftarrow \text{argmax}(\ell_{0\text{-}1} \text{ reduction})$
44:             $\mathcal{T} \leftarrow \mathcal{T} - [l]$
45:             Append $\mathcal{T}$ to $\mathbb{T}$
46:         **end while**
47:         $\mathcal{T} \leftarrow \text{argmin}_{\mathcal{T} \in \mathbb{T}}(\ell_{0\text{-}1} \text{ plus } \alpha\text{-scaled tree size})$
48:     **end if**
49: **end for**

---

# B    COMPARISON TO MODEL-FREE REINFORCEMENT LEARNING

One of the most consistently observed benefits of model-based RL is its sample efficiency, and this trend holds in our context. Running Algorithm 1 unchanged except for the use of a soft actor-critic (SAC) agent for policy learning, we find that approximately two orders of magnitude more environment interaction is required to achieve equivalent performance in terms of regret at convergence. In turn, this increases wall-clock runtime by 10-20 times, thereby outweighing the higher per-timestep computational cost of PETS over SAC. The caption of Figure A2 gives further details.

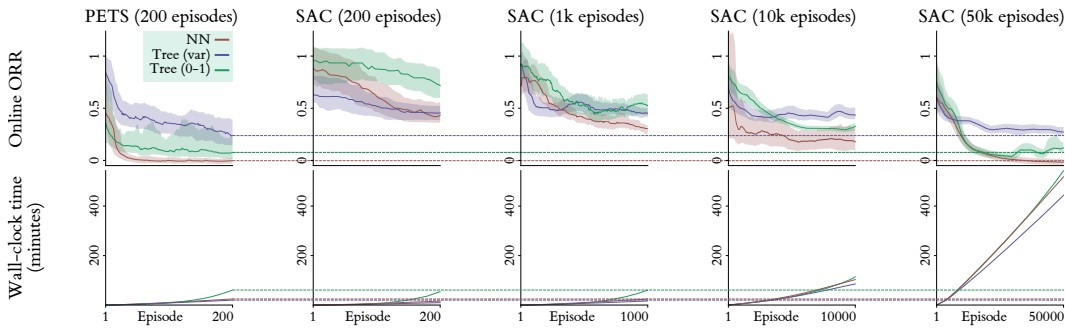

Figure A2: Comparing the use of PETS (model-based) and SAC (model-free) agents on the Follow task (see Appendix C.3 for task details). The PETS results are taken directly from Figure 3. For SAC we retain the total preference budget of $K_{max} = 1000$, but for longer runs add episode trajectories to $\Xi$ and $\mathcal{L}$ at a reduced frequency so that $N_{max} = 200$ (e.g. for 50000 episodes, only 1 in 250 episodes are added to the graph; the rest are skipped). All SAC agents use policy and value networks with two 256-unit hidden layers each, learning rates of $1e^{-4}$ and $1e^{-3}$ for the policy and value updates with the Adam optimiser, a discount factor of $\gamma = 0.99$, and an interpolation factor of 0.99 for Polyak averaging of the target networks. Updates use mini-batches of 32 transitions sampled uniformly from a rolling replay buffer of capacity $5e^4$.

We find that initially running SAC for a total of 200 episodes, matching our PETS experiments, gives the model-free learning algorithm insufficient time to achieve good performance in terms of regret on the oracle reward function (a higher learning rate leads SAC to become unstable). We then progressively increase the length of SAC runs until regret performance matches the use of PETS, and find that this requires around 50000 episodes, an increase of 250 times. It is noteworthy that the variance-based tree model seems to perform best in the short-runtime regime, but worst in the long-runtime regime. Time constraints prevented us from investigating whether this holds in other tasks, but such an investigation would be worthwhile.

In terms of wall-clock time (on a single NVIDIA Tesla P100 GPU), running reward learning with SAC for 1000 episodes is roughly equivalent to 200 episodes using PETS (25-60 minutes, depending on the reward model architecture). For 50000 episodes, this time increases to 9 hours. This brings a very practical disadvantage: if reward learning were done using human preferences instead of an oracle, that person would have to dedicate more than a full working day to the exercise, most of which would be spent waiting for several minutes between each successive preference batch.

*Note: The PETS wall-clock times quoted here exclude the time to pre-train the dynamics models. Although this is not how MBRL sample complexity is typically measured, we argue that it is appropriate for the reward learning context, where the key factor is the period for which a human would be required to be in-the-loop. Regardless, pre-training in the aircraft handling domain takes around 30 minutes, which remains low compared with the 9 hours for high-performing SAC agents.*

## C AIRCRAFT HANDLING ENVIRONMENT

### C.1 MOTIVATION

Pilots of fast jet aircraft require exceptional handling abilities, acquired over years of advanced training. There would be immense practical value in developing a method for distilling the knowledge and preferences of pilots and other domain experts into a software model that captures realistic handling behaviour. The scalability of such a model would make it useful for strategic planning exercises, training of a range of operational roles, and development and testing of other software systems. However, as in many contexts where intuitive decision-making and rapid motor control are paramount, the preferences of experts (over the space of fast jet handling trajectories) are in large part tacit, and thus defy direct scrutiny or verbal description. Put simply: experts know good handling when they see it, but cannot directly express *why*.[1] This makes it practically challenging to accurately elicit this knowledge for codification into an automated system.

The methods presented in this paper form the basis of a possible solution to this dilemma. Given a dataset of trajectories executed by an artificial learning agent and labelled with pairwise expert preferences (which require only tacit knowledge to produce), we use statistical learning algorithms to construct an interpretable explanatory model of those preferences. The result is two distinct outputs that could form valuable components of future planning, training and development software:

1. A tree-structured reward function, which may be used for automated scoring of flight trajectories executed by human or artificial pilots. We aim for this to produce an evaluation that is consistent, unbiased and aligned with the judgement that the original expert would have made, alongside an explanatory rationale that can be leveraged to justify, verify and improve handling behaviour.
2. A model-based RL agent capable of executing high-quality handling behaviour with respect to the reward function, for use in simulation.

It should be noted that any realistic handling scenario would involve multiple experts somewhat-differing knowledge and expertise. A natural extension of our approach, which we see as valuable future work, is to learn individual reward functions for each expert, then leverage the intrinsic interpretability to identify biases, inconsistencies and tradeoffs. This suggests a third application of reward tree learning: providing a basis for evaluating and training the experts themselves.

### C.2 IMPLEMENTATION

We consider a simple set-piece formulation of the aircraft handling problem, in which the piloting agent is given a short time window to manoeuvre their aircraft (the ego jet, EJ) in a particular manner relative to a second reference jet (RJ). Special cases of this formulation create a wide variety of tasks for the pilot to solve. Options include:

- RJ is a *friendly* aircraft which EJ should accompany in formation flight.
- RJ is *adversarial* and EJ must outmanoeuvre it to gain a tactical advantage.
- Rather than being a distinct physical entity, RJ defines a *goal pose* (position and attitude) for EJ to reach. The goal pose may be fixed or moving over time.

We developed this formulation to strike a balance between simplicity and generality; many realistic scenarios faced by a fast jet pilot involve interaction with a single other airborne entity. On a practical level, it provides scope for the definition of many alternative tasks given the same state and action spaces, and largely unchanged dynamics.

The state space contains the positions, attitudes, velocities and accelerations of both EJ and RJ (state dimensionality $= 37$) and the action space consists of pitch, roll, yaw and thrust demands for

---

[1]This statement certainly underestimates the rich complexity of human expertise; in reality, an expert's mental model is likely to be partly tacit and partly explicit. The general strategy of preference-based reward learning is to operate *as if* the mental model were $100\%$ tacit, and explore what can be achieved under such a strong restriction. Real-world applications would likely benefit from combining this approach with some amount of hand-coded expert knowledge.

EJ only (action dimensionality $= 4$). The EJ dynamics function integrates these demands with a simplified physics engine, including gravity and air resistance (we make no claim of realism here; the simulator is merely a proof of concept). A new action is accepted every 25 steps of the physics engine, reducing an agent's decision frequency to approximately 1Hz. RJ dynamics, as well as the conditions of state initialisation and termination, vary between tasks (see Appendix C.3).

The final generic aspect of the implementation is the feature function $\phi$, which maps the transition space $\mathcal{S} \times \mathcal{A} \times \mathcal{S}$ (total dimensionality $= 37 + 4 + 37 = 78$) into an $F$-dimensional space of task-relevant features. In consultation with engineers with experience of aerospace simulation and control algorithms, we devised the following set of $F = 30$ features that is sufficiently expressive to capture the important information for all three of our target tasks, without being overly specialised to one or providing too much explicit guidance to the reward learning process. Apart from those containing "delta" or "rate", all features are computed over the successor state for each transition, $s_{t+1}$.

| | |
|---|---|
| dist | Euclidean distance between EJ and RJ |
| closing speed | Closing speed between EJ and RJ (negative = moving closer) |
| alt | Altitude of EJ |
| alt error | Difference in altitude between EJ and RJ (negative = EJ is lower) |
| delta alt error | Change in alt error between $s_t$ and $s_{t+1}$ |
| dist hor | Euclidean distance between EJ and RJ in horizontal plane |
| delta dist hor | Change in dist hor between $s_t$ and $s_{t+1}$ (negative = moving closer) |
| pitch error | Absolute difference in pitch angle between EJ and RJ |
| delta pitch error | Change in pitch error between $s_t$ and $s_{t+1}$ |
| abs roll | Absolute roll angle of EJ |
| roll error | Absolute difference in roll angle between EJ and RJ |
| delta roll error | Change in roll error between $s_t$ and $s_{t+1}$ |
| hdg error | Absolute difference in heading angle between EJ and RJ |
| delta hdg error | Change in hdg error between $s_t$ and $s_{t+1}$ |
| fwd error | Angle between 3D vectors indicating forward axes of EJ and RJ |
| delta fwd error | Change in fwd error between $s_t$ and $s_{t+1}$ |
| up error | Angle between 3D vectors indicating upward axes of EJ and RJ |
| delta up error | Change in up error between $s_t$ and $s_{t+1}$ |
| right error | Angle between 3D vectors indicating rightward axes of EJ and RJ |
| delta right error | Change in right error between $s_t$ and $s_{t+1}$ |
| los error | Angle between forward axis of EJ and vector from EJ to RJ (measures whether RJ is in EJ's line of sight) |
| delta los error | Change in los error between $s_t$ and $s_{t+1}$ |
| abs lr offset | Magnitude of projection of vector from EJ to RJ onto RJ's rightward axis (measures left-right offset between the two aircraft in RJ's reference frame) |
| speed | Airspeed of EJ |
| g force | Instantaneous g-force experienced by EJ |
| pitch rate | Absolute change of EJ pitch between $s_t$ and $s_{t+1}$ |
| roll rate | Absolute change of EJ roll between $s_t$ and $s_{t+1}$ |
| yaw rate | Absolute change of EJ yaw between $s_t$ and $s_{t+1}$ |
| thrust | Instantaneous thrust output by EJ engines |
| delta thrust | Absolute change in thrust between $s_t$ and $s_{t+1}$ |

## C.3 Tasks and Oracles

In this paper, we consider three concrete tasks that instantiate the general EJ-RJ framework. For each, we construct a plausible oracle reward function from a subset of the 30 features, meaning that reward learning is in part a feature selection problem (tree models perform feature selection explicitly whenever they add a new splitting rule). Although the precise nature of the oracle reward functions is secondary, and those given below are among many equally reasonable alternatives, we dedicated several hours of development time to ensuring they yield reasonable behaviour upon visual inspection. The difficulty and seeming arbitrariness of this manual reward design process is precisely why reward learning (ultimately from real human preferences) is an enticing proposition. Descriptions of the three tasks, along with their respective oracles, are given below:

- **Follow**: Here RJ follows a linear horizontal flight path at a constant velocity, which is oriented opposite to the initial velocity of EJ. The goal is to turn onto and then maintain the path up to the episode time limit of 20 timesteps. This constitutes a very simple form of

formation flight. The oracle reward function incentivises closing the distance to the moving target, and matching the upward axes of EJ and RJ:

$$r = -(\texttt{dist} + 0.05 \times \texttt{closing speed} + 10 \times \texttt{up error}).$$

- **Chase**: Here RJ follows an erratic trajectory generated by random control inputs, and the goal is to chase it without taking EJ below a safe altitude of 50. Episodes terminate after 20 timesteps. The oracle reward function incentivises keeping RJ at a distance of 20 and within EJ's line of sight, while keeping EJ oriented upright. It also has a large penalty for dropping below the safe altitude:

$$r = -\left(\text{abs}(\texttt{dist} - 20) + 10 \times \texttt{los error} + 5 \times \texttt{abs roll} + \begin{cases} 100 & \text{if } \texttt{alt} < 50 \\ 0 & \text{otherwise} \end{cases}\right).$$

- **Land**: Here the goal is to execute a safe approach towards landing on a runway, where RJ represents the ideal landing position (central, zero altitude, slight upward pitch). EJ is initialised at a random altitude, pitch, roll and offset, such that landing may be challenging but always physically possible. An episode terminates if EJ passes RJ along the axis of the runway, or after 25 timesteps otherwise. The oracle reward function for this task is by far the most complex of the three, including terms that incentivise continual descent, penalise g-force and engine thrust, and punish the agent for making contact with the ground ($\texttt{alt} < 0.6$) before the start of the runway:

$$r = -(0.05 \times \texttt{abs lr offset} + 0.05 \times \texttt{alt} + \texttt{hdg error} + \texttt{abs roll}$$
$$+ 0.5 \times \texttt{pitch error} + 0.25 \times (\texttt{yaw rate} + \texttt{roll rate} + \texttt{pitch rate})$$
$$+ 0.1 \times \texttt{g force} + 0.025 \times \texttt{thrust} + 0.05 \times \texttt{delta thrust}$$
$$+ \begin{cases} 1 & \text{if } \texttt{delta dist hor} > 0 \\ 0 & \text{otherwise} \end{cases} + \begin{cases} 2 & \text{if } \texttt{delta alt} > 0 \\ 0 & \text{otherwise} \end{cases}$$
$$+ \begin{cases} 1 & \text{if } \texttt{abs lr offset} > 10 \\ 0 & \text{otherwise} \end{cases} + \begin{cases} 10 & \text{if } \texttt{alt} < 0.6 \\ 0 & \text{otherwise} \end{cases}).$$

## D    IMPLEMENTATION AND EXPERIMENT DETAILS

### D.1    ORACLE PREFERENCES

Oracle preferences are generated in accordance with the Bradley-Terry model given in Equation 1, i.e. by computing the returns for the two trajectories $\xi^i$ and $\xi^j$, and sampling from a Boltzmann distribution parameterised by those returns. In our main experiments, we set the temperature coefficient $\beta = 0$, which results in the oracle deterministically selecting the trajectory with higher return (ties broken uniform-randomly). In Section 6.3 we study cases with $\beta > 0$, which provide a more realistic emulation of real human preference data.

### D.2    HYPERPARAMETERS FOR TREE INDUCTION

In all experiments, we use the following hyperparameters during tree induction. These were identified through informal search, and we make no claim of optimality, but they do lead to reasonable performance on the three tasks of varying complexity. This indicates a general insensitivity of the method to precise hyperparameter values, which is often practically advantageous.

- Trajectory return estimation using the Adam optimiser with a learning rate of $0.1$. Optimisation stops when the mean $\ell_{\mathrm{NLL}}$ changes by $< 1e^{-5}$ between successive gradient steps.

- Per-feature candidate split thresholds $\mathcal{C}$ defined as all midpoints between adjacent unique values in the trajectory set $\Xi$. These are recomputed on each update.

- Tree size limit $L_{\max} = 100$.

- Tree size regularisation coefficient $\alpha = 5e^{-3}$.

As mentioned in Appendix A.2, we enforce negative rewards (max $= 0$ constraint) for the Land task, and positive rewards (min $= 0$ constraint) for Follow and Chase.

### D.3    MODEL-BASED RL IMPLEMENTATION

For conceptual details on the PETS algorithm, we refer readers to the original paper by Chua et al. (2018). In our implementation, the dynamics model is an ensemble of five NNs, each with four hidden layers of 200 hidden units and ReLU activations. State vectors are pre-normalised by applying a hand-specifed scale factor to each dimension. Decision-time planning operates over a time horizon of $H = 10$ and consists of 10 iterations of the cross-entropy method. Each iteration samples 20 candidate action sequences from an independent Gaussian, of which the top 5 in terms of return are identified as *elites*, then updates the sampling Gaussian towards the elites with a learning rate of $0.5$. In all experiments we use $\gamma = 1$, meaning no temporal discounting is applied during planning.

In our experiments, we find that the particular dynamics of the aircraft handling environment permit us to pre-train $D'$ on random offline data, and accurately generalise to states encountered during online reward learning. This means we perform no further updates to the model while reward learning is ongoing. As well as improving wall-clock speed, this avoids complexity and convergence issues arising from having two interacting learning processes (note that simultaneous learning is completely unavoidable with model-free RL). To pre-train, we collect $1e^5$ transitions by rolling out a uniform random policy, then update each of the five networks on $1e^5$ independently sampled mini-batches of 256 transitions, using the mean squared error loss over normalised next-state predictions.

### D.4    NEURAL NETWORK REWARD LEARNING BASELINE

We baseline our reward tree models against the de facto standard approach of reward learning using a NN. In constructing this baseline, we aimed to retain as much of Algorithm 1 as possible, so that only the model architecture varies. The result is that we replace lines 21-47 with the following:

---

21: **for** $m \in \{1, ..., M\}$ **do**
22: $\quad \mathcal{L}_{\text{mini-batch}} \leftarrow$ a mini-batch of $B$ preference labels sampled from $\mathcal{L}$
23: $\quad$ Compute $\ell_{\text{NLL}}$ over $\mathcal{L}_{\text{mini-batch}}$ via Equations 1 and 2
24: $\quad$ Backpropagate loss and update network parameters
25: **end for**
26: $\mathbf{r}_{\text{all}} \leftarrow$ reward predictions for all feature vectors in $\Xi$
27: Scale network outputs by $1/\text{std}(\mathbf{r}_{\text{all}})$
28: Shift network outputs by $-\min(\mathbf{r}_{\text{all}})$ or $-\max(\mathbf{r}_{\text{all}})$, depending on desired reward sign

---

The new lines 26-28 replicate the two constraints applied in Equation 3.

In all experiments, we follow Lee et al. (2021b) in implementing the reward model as a three-layer network with 256 hidden units each and leaky ReLU activations, and performing the update on line 24 using the Adam optimiser (Kingma & Ba, 2014) with a learning rate of $3e^{-4}$. On each update, we sample $M = 100$ mini-batches of size $B = 32$ and take one gradient step per mini-batch.

## D.5 COVERAGE DATASETS FOR POLICY-INVARIANT EVALUATION

Gleave et al. (2021) recently highlighted the importance of comparing and evaluating learnt reward functions in a policy-invariant manner, by using a common evaluation distribution rather than on-policy data generated by agents optimising for each reward. Ideally, the offline evaluation data should have high coverage (i.e. high-entropy state distribution, both high- and low-quality trajectories), in order to characterise the reward functions' outputs across a spectrum of plausible policies.

In our context, we can generate data that satisfies these requirements by leveraging the known oracle reward functions and the PETS algorithm. We deploy PETS using the oracle reward, but randomise the planning parameters (number of planning iterations $\in \{1, ..., 50\}$, number of action sequence samples $\in \{4, ..., 50\}$) on every episode. In all cases, we take the top $25\%$ of action sequences as elites. This randomisation results in trajectories that are sometimes near-optimal with respect to the oracle, sometimes moderate in quality, and sometimes barely better than random. For all three tasks, we generate a dataset of 200 evaluation trajectories in this manner.

# E    VISUALING REWARD FUNCTIONS WITH SIMILARITY EMBEDDINGS

Here we briefly discuss a novel form of reward visualisation which we developed and found valuable during our own analysis. Given some measure of similarity between reward functions, such as the EPIC metric proposed by Gleave et al. (2021), we can compute a matrix of pairwise similarities between any number of such functions (computational cost permitting). We can then produce a 2D embedding of the functions by applying multidimensional scaling (MDS). Visualising this embedding as a scatter plot enables the discovery of salient patterns and trends in the set of functions.

In Figure A3, we use the metric of rank correlation on the coverage datasets, and the SMACOF MDS algorithm (De Leeuw & Mair, 2009), to embed all 30 model repeats and the oracle for each task. This gives an impression of the models' similarity not just to the oracle, but to each other. Aside from the Follow NNs, which form a tight cluster near the oracle, the distribution for each model indicates roughly equal consistency between repeats. The overlap of convex hulls suggests that the rankings produced by all models are broadly similar for Land, but more distinct for Chase. Shading points by ORR reveals that while models further from the oracle tend to induce worse-performing policies, the trend is not monotonic. This reinforces the point made elsewhere that the problems of learning a good policy and exactly replicating the ground truth reward are not identical.

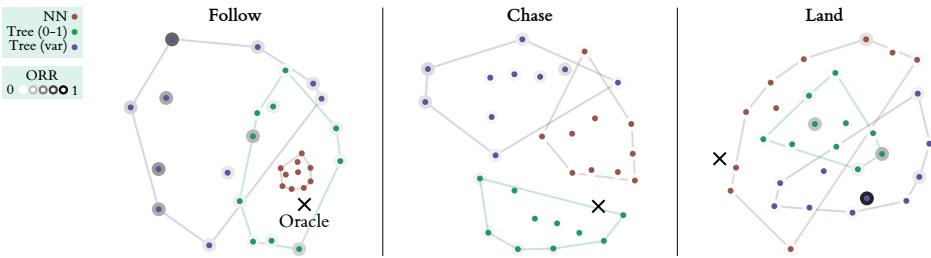

Figure A3: Rank correlation embeddings for all model repeats from the main experiments, with the scatter point for each repeat shaded by ORR.

Populating such embedding plots more densely, perhaps by varying model hyperparameters, could provide a means of mapping the space of learnable reward functions and its relationship to policy performance. It would also be straightforward to compute similarity values for the same model repeat at multiple checkpoints during learning. This would yield a trajectory in the rank embedding space, which could aid the assessment of the stability and convergence properties of online learning with different models and hyperparameter values.

