# OpenReview forum: "Reward Learning with Trees: Methods and Evaluation"
_ICLR.cc/2023/Conference — Submitted to ICLR 2023_

### Official Review · Reviewer_1NsX · 2022-10-15

**Confidence:** 4
**Correctness:** 3
**Technical Novelty And Significance:** 2
**Empirical Novelty And Significance:** 2
**Recommendation:** 3

**Clarity, Quality, Novelty And Reproducibility:**

As I mentioned above, the paper is well-structured and easy to follow, but the proposed updates on the reward tree are incremental and the novelty is not well-defined. In terms of reproducibility, I have not run the code, but after scanning through the appendix, I believe the paper has sufficient details for supporting reproduction.

**Strength And Weaknesses:**

**Strength**:
1. The paper is well-structured and easy to follow. The reward tree is clearly defined with sufficient details (in both the main paper and appendix) for supporting implementation and reproduction.
2. The empirical results demonstrate the performance (including fidelity and interpretability) of the reward tree from multiple perspectives, although the paper studies only an aircraft handling environment which is not a common benchmark for RL.


**Weaknesses**:

1. **Contribution.** The novelty of this paper is marginal. The main methodological contribution is the reward tree, but this method was proposed by a previous paper and it is well-studied under the RL environment. The updates on the reward tree are incremental, without strong motivations or careful justification about why these updates are critical. The main argument is that the performance can be somehow improved with these updates, but this argument is fairly weak to be a part of a scientific paper, especially for the papers to be published at top-tier conferences such as ICLR. I strongly recommend the author(s) to focus on one of these updates and justify the necessity of this update with theory results and relevant experiments.

2. **Evaluation Environment** This paper focuses only on an aircraft handling environment, but this environment is not a common benchmark for RL. I do recommend using some popular environments like Atari and MoJoCo. Most of the RL works are evaluated in these environments, and in this way, readers can understand how well the proposed method advances the relevant research fields/topics and whether this method can be scaled to popular RL tasks.

3. **Related Works** I believe Inverse Reinforcement Learning (IRL) shares the same goal as this paper. The major difference is IRL does not require human feedback while the proposed task has to rely on such signals during reward inference. Whether this human-in-the-loop system can be generalized to solve complex tasks that require a large number of interactions and long-term planning is questionable. I won't argue for this issue since I understand the authors just follow some previous works that have the same issue, but I believe the author should at least mention the works of IRL in the paper.

4. **Additional Concerns**

- "*We show it to be broadly competitive with neural networks on challenging high-dimensional tasks,*"
By claiming high-dimensional tasks in RL, we often refer to image-based games (e.g., Atari) or text-based games (e.g., TextWorld). Recently year, Deep RL has achieved significant performance in these games while the tree model can not. I think the current reward tree cannot be scaled to these tasks, but it could be a promising direction to explore or advance.

- "*These parameters are non-differentiable, making end-to-end optimization of the losses in Equation 2 computationally intractable.*"
This is a common issue for the tree model, but I have not found the corresponding solution or circumvention. I guess the issue remains unsolved, but In fact, many differentiable tree models have been proposed in recent years and I do recommend referring to these works.

- "*For consistency with prior work, we instead minimize the NLL loss under the Bradley-Terry model.*"
I believe "consistency with prior work" is the motivation for the NLL loss. However, it is a well-known fact that the square loss can be consistent with the log-likelihood loss if we assume the targets (or returns in this work) are Gaussian distributed with unit variance. This motivation needs to be clarified.

- "*We optimise Equation 3 by unconstrained gradient descent with the Adam optimiser (Kingma & Ba, 2014), followed by post-normalisation to meet the constraints*"
I recommend expanding this sentence by defining the post-normalization method. Have you projected the constrained objective to the unconstrained one in each step? If yes, which projection framework have you applied? I cannot imagine how post-normalization can achieve this projection. I question the correctness here, please explain.

- "*we find that switching from SAC to a model-based algorithm called PETS (Chua et al., 2018) reduces training steps by multiple orders of magnitude, and cuts wall-clock runtime (see Appendix B).*"
One of the main challenges of MBRL is learning the transition dynamics based on collected samples, and we often count the training steps for the transition model when we compare  MBRL's sample complexity with model-free methods. It seems in this paper the comparison is based on a well-trained transition model if I understand correctly, which makes the conclusion questionable.

**Summary Of The Paper:**

This paper introduces the method of learning interpretable trees for representing reward functions by following a previously-proposed reward tree model. The paper proposes several updates to the reward tree inference algorithm, including an NLL-based objective for estimating trajectory-level returns, a new criterion facilitating tree growth, and a model-based RL method. This paper empirically studies the performance of their model on an aircraft handling environment by mainly focusing on three control tasks, including 'Follow', 'Chase' and 'Land'. The results demonstrate the performance of the proposed updates from the perspectives of quantitative performance, visual inspection, sensitivity analysis, and tree structure analysis.

**Summary Of The Review:**

The authors spend considerable effort to demonstrate the performance of reward trees, which is remarkable in my mind. However, the reward tree has been proposed in previous work, and the updates mentioned in this paper are marginal and incremental without enough evidence to justify why they are critical. I also have some concerns about the argument made in the paper, and I hope the authors can respond to them. The environment picked for the empirical study is not satisfying. The author should consider including some more popular environments. I think I have provided some useful suggestions which could significantly improve the paper (I do believe so), but implementing them takes some time and effort. In conclusion, I vote for a rejection based on the current version.

---

> ### Author Response · Authors · 2022-11-11
> **Response to Reviewer 1NsX (Part 1)**
>
> Dear Reviewer 1NsX, thanks for taking the time to post your review!
>
> **As a first port of call, we invite you to read our two-part global response, which addresses concerns raised by all three reviewers.**
>
> Below are responses to additional points made by yourself:
> - *"The main methodological contribution is the reward tree, but this method was proposed by a previous paper and it is well-studied under the RL environment"* - Our global response covers the general issue of the methodological contribution. We politely disagree with your assertion that reward trees are *"well-studied"*; prior to the single existing work by Bewley & Lecue (BL), trees had never been used for reward learning. They have seen use in other parts of the RL pipeline (e.g. policy, value function), but these applications have very different constraints and challenges.
> - *"The main argument is that the performance can be somehow improved with these updates, but this argument is fairly weak... I strongly recommend the author(s) to focus on one of these updates and justify the necessity of this update"* - Our paper already does exactly what you suggest. In our experiments, we focus on the change in split criterion from variance to the 0-1 loss, and find a significant gain in quantitative and qualitative performance. We therefore contend that we have presented a rather strong empirical argument for implementing this update.
> - *"I believe Inverse Reinforcement Learning (IRL) shares the same goal as this paper"* - It seems your critique is in this paragraph concerns preference-based reward learning in general. The preference-based and IRL approaches have complementary merits and drawbacks, as has been acknowledged in numerous prior works [1,2,3]. There is no reason why the two could not be combined for reward tree learning in future, and we think this would be an excellent area for further work. You say that we should *"should at least mention the works of IRL"*, but we do just that in the related work section by citing the seminal work by Ng and Russell [4]. The confusion may be that we refer to the technique not as *"inverse reinforcement learning"*, but as *"reward learning from demonstrations"*; the two terms are commonly used to mean the same thing [1,5].
>
> [1] Ibarz, B., Leike, J., Pohlen, T., Irving, G., Legg, S. and Amodei, D., 2018. Reward learning from human preferences and demonstrations in atari. Advances in neural information processing systems, 31.
>
> [2] Palan, M., Shevchuk, G., Charles Landolfi, N. and Sadigh, D., 2019, June. Learning Reward Functions by Integrating Human Demonstrations and Preferences. In Robotics: Science and Systems.
>
> [3] Bıyık, E., Losey, D.P., Palan, M., Landolfi, N.C., Shevchuk, G. and Sadigh, D., 2022. Learning reward functions from diverse sources of human feedback: Optimally integrating demonstrations and preferences. The International Journal of Robotics Research, 41(1), pp.45-67.
>
> [4] Ng, A.Y. and Russell, S., 2000, June. Algorithms for inverse reinforcement learning. In ICML (Vol. 1, p. 2).
>
> [5] Michini, B., Cutler, M. and How, J.P., 2013, May. Scalable reward learning from demonstration. In 2013 IEEE International Conference on Robotics and Automation (pp. 303-308). IEEE.
>
> [Continued in Part 2...]

---

> > ### Author Response · Authors · 2022-11-11
> > **Response to Reviewer 1NsX (Part 2)**
> >
> > [...Continuation from Part 1]
> >
> > - *"By claiming high-dimensional tasks in RL, we often refer to image-based games... I think the current reward tree cannot be scaled to these tasks"* - While it is not fruitful to argue over semantics, we feel the term *"high-dimensional"* is also applicable to continuous state spaces with dozens of dimensions. We disagree that reward trees are fundamentally incompatible with image-based environments, but they would need to be complemented by either (a) the manual specification of a reward-relevant feature vector or (b) an initial stage of (interpretable) state representation learning [6]. As you say, this would be *"a promising direction to explore or advance"*.
> > - *"This [non-differentiability] is a common issue for the tree model, but I have not found the corresponding solution or circumvention"* - The multi-stage model induction process outlined from that point onwards *is* the solution/circumvention of the non-differentiability problem. It relies on first constructing trajectory-level return estimates, then growing a tree while keeping those estimates fixed.
> > - *"Many differentiable tree models have been proposed in recent years and I do recommend referring to these works"* - This is a valid point, and we will add a citation to a differentiable tree model. One reason why we did not consider using a differentiable tree in this work is that, to the best of our knowledge, all such models learn oblique (c.f. axis-aligned) splits, whose interpretability in high-dimensional feature spaces is rather questionable. **If you know of any differentiable models for axis-aligned tree learning, we would be very interested to hear about them!** We restricted ourselves to the less expressive but more interpretable class of axis-aligned trees, and found that they still performed admirably in comparison to neural networks.
> > - *"It is a well-known fact that the square loss can be consistent with the log-likelihood loss if we assume the targets (or returns in this work) are Gaussian distributed with unit variance"* - Our regression targets are not the returns, but rather the (binary) pairwise preference labels, so the NLL and square loss are definitely not consistent in our case. That said, your confusion is valid as we should have been more precise with our description of the loss functions. We will work to find a better wording.
> > - *"I recommend expanding this sentence by defining the post-normalization method... I question the correctness here, please explain"* - The method is very simple; we just freely optimise the NLL loss by gradient descent until convergence, then apply a shift and scale factor afterwards so that the return estimates end up with a common sign and a standard deviation of $\beta$. We will again seek to improve the wording here.
> > - *"We often count the training steps for the transition model when we compare MBRL's sample complexity with model-free methods"* - This is true, and our current phrasing is perhaps misleading. What we mean to say here is that the step count and wall-clock time *during reward learning* is greatly reduced, which is important because that is the period when a human would be required to be in-the-loop. Front-loading the sample complexity and wall-clock time onto the dynamics learning phase is likely to reduce the amount of "dead time" for which that person would need to wait around between preference batches. In addition, the same dynamics model may be reusable for multiple runs of reward learning with different human evaluators (assuming no major distributional shift), meaning the effective sample complexity of each run is reduced yet further. Once again, we will revisit our choice of wording.
> >
> > [6] Lesort, T., Díaz-Rodríguez, N., Goudou, J.F. and Filliat, D., 2018. State representation learning for control: An overview. Neural Networks, 108, pp.379-392.

---

> > > ### Comment · Reviewer_1NsX · 2022-11-19
> > > **Acknowledging the response and more comments**
> > >
> > > Thanks for clarifying. After going over the response, I am still feeling that most of my arguments are valid. I can provide some follow-up comments, but my rating remains the same. The authors make lots of **strong claims**, but the evidence supporting their claims is limited.  I suggest considering my comment in the decision, but I won't argue for a firm acceptance or rejection. I leave the decision to our area chair.
> > >
> > > *"we focus on the change in split criterion from variance to the 0-1 loss"*
> > > The first appearance of the term 0-1 loss is on Page 3 (formulate 2). Neither the abstract nor the introduction has mentioned it. I can't entirely agree the paper has primarily focused on this update.
> > >
> > > *"The preference-based and IRL approaches have complementary merits and drawbacks, as has been acknowledged in numerous prior works [1,2,3]. There is no reason why the two could not be combined for reward tree learning in the future"*
> > > I cannot agree that IRL and the reward tree are complementary to each other but rather they are comparable to each other. They share the same goal, the only difference is whether human interaction is necessary during learning.
> > >
> > > *"Our regression targets are not the returns, but rather the (binary) pairwise preference labels"*
> > > Using a different label won't influence the fact that square loss and LL loss are similar. Replacing one with another cannot be a real contribution in my mind.
> > >
> > > *"Can reward trees learnt from preferences perform competitively with neural networks (NNs) in a realistic and industrially-relevant domain, while remaining interpretable?"*
> > > How about the image, video, and word inputs? The major advantages of NN are based on these inputs. I cannot agree a tree model can replace NN without some strong extension, and by the way, I have not seen any industrial application that has been studied in this paper.

---

> > > > ### Author Response · Authors · 2022-11-19
> > > > **Responses to your follow-up comments**
> > > >
> > > > Thanks for your additional comments! Further clarification below:
> > > >
> > > > *The first appearance of the term 0-1 loss is on Page 3 (formulate 2). Neither the abstract nor the introduction has mentioned it. I can't entirely agree the paper has primarily focused on this update.*
> > > >
> > > > By "focus on" here we refer to the fact that our **evaluation** considers the effect of making this change - 0-1-based splitting trees are baselined against BL's variance-based method throughout. We are not claiming that **the paper as a whole** is primarily focused on this, or any other, methodological change. As outlined in our Global Response (Part 1), the main focus of the paper is on the finding that reward trees as a general method can be competitive with NNs in a realistic and industrially-relevant domain, while remaining interpretable. Methodological changes are presented as a secondary contribution.
> > > >
> > > > *I cannot agree that IRL and the reward tree are complementary to each other but rather they are comparable to each other. They share the same goal, the only difference is whether human interaction is necessary during learning.*
> > > >
> > > > With respect, we suggest you might be conflating two concepts: the reward learning *information source* (preferences vs demonstrations, with the latter being IRL) and the *model architecture* used to perform that learning (neural network vs tree). In our paper, we have demonstrated that trees can be used instead of neural networks to learn reward functions from preferences. Our suggestion is that algorithms could be developed to learn reward trees from demonstrations instead. In fact, we suspect that many of the model induction steps would remain the same.
> > > >
> > > > *Using a different label won't influence the fact that square loss and LL loss are similar. Replacing one with another cannot be a real contribution in my mind.*
> > > >
> > > > To your first sentence:
> > > > - We clarified that the regression targets are binary in direct response to your point that the square and NLL losses are equivalent *"if we assume the targets... are Gaussian distributed"*. Binary variables (discrete) cannot be Gaussian distributed (continuous), so the premise that the two losses might be equivalent doesn't hold up.
> > > >
> > > > To your second sentence:
> > > >
> > > > - We agree, and as we make clear in the paper, the reason for our change in loss function is for *consistency* with prior work (i.e. the direct opposite of novelty!) BL's use of the square loss has far less precedence, and is less well theoretically grounded in the reward learning context.
> > > >
> > > > *How about the image, video, and word inputs? The major advantages of NN are based on these inputs. I cannot agree a tree model can replace NN without some strong extension.*
> > > >
> > > > We're not sure we agree with the blanket statement that *"the major advantages of NN are based on these inputs"*. NNs are so successful because they are flexible function approximators across a wide range of input domains, including continuous ones. Our paper has shown that another, more interpretable function approximator -- a tree model -- can perform almost as well as NNs in the context of reward learning from trajectory preferences. Adapting reward trees for image, video, and word inputs would indeed require a strong extension, but this does not mean it is worth studying.
> > > >
> > > > *By the way, I have not seen any industrial application that has been studied in this paper.*
> > > >
> > > > Our aircraft handling experiments developed through consultation with an industrial partner with interest in precisely this use case, and Appendix C.1 discusses why it is interesting and valuable. We believe it is reasonable to say that this domain is more industrially relevant than, for example, Atari games, for which no obvious connection to an industrial application exists.

---

### Official Review · Reviewer_aaeK · 2022-10-19

**Confidence:** 4
**Correctness:** 3
**Technical Novelty And Significance:** 2
**Empirical Novelty And Significance:** 3
**Recommendation:** 6

**Clarity, Quality, Novelty And Reproducibility:**

As indicated earlier, the clarity and quality of the approach are co-dependent: the paper is solid, and the authors do quite a good job in explaining all aspects (the paper is overall well written) but it can be improved. Novelty is a weak point as discussed earlier. Reproducibility should not be an issue if code would be provided, and based on the description a lot is possible too, but with some additional intepretation/implementation or help.

**Strength And Weaknesses:**

The strengths of this paper are
1) it is a solid paper in terms of the text, the structure, the content,
2) it is a solid set of experiments with clear goals, good methods, and extensive analysis,
3) the topic is interesting as explainable models that can rival the (still) less explainable neural networks for RL tasks are interesting,
4) the description of the method, including some tweaks and small extensions are solid.

The main weaknesses of the paper are:
5) the novelty and significance are limited, especially compared to the state-of-the-art in general and the work by Bewley and Lecue (2022), BL, in particular.
6) The paper is very dense, which is a good thing because it is quite informative for that matter, but for some parts I feel that more general intuitions would be better in the main tet than highly detailed descriptions of all outcomes (I mean, there are still appendices).
7) The experimental evaluation is extensive, but has a very simple conclusion: NNs are still often better (in many ways) as expected because of the more general model class, but tree structures can help in explanations, which is not new at all.
I'll elaborate a bit more on 5-7:

About (5) I feel that the extension of BL is only minor. I checked the original BL paper for that, and the description is different in some ways (notationally) but overall it seems that the main difference lies in somewhat different optimization in the first step and a different criterion in the third step. This difference is not analysed/evaluated later on anyway. Another difference is the use of the method in model-based RL settings, whereas BL focuses on model-free settings. The current paper uses a very naive approach where both tree induction and planning happens constantly from scratch, hence also here no real new mechanisms are in place. I think it would be good to highlight the algorithmic and performance-wise aspects of these parts more explicitly in a kind of contributions, to make this more clear.
This could also be said about the related work section: it is nice and informative, and certainly substantial, but I do miss these explicit placements of the current method in the state-of-the-art. Also, I think that "trees for explainable" is something that has been considered before in RL for decades, although not explicitly so in terms of "explainable" since that is a more recent development. Tree-based models in RL have a long history, and it would be good to connect to much more older work to, to highlight that. Take, for example, the seminal work by Dzeroski, Blockeel and De Raedt on relational RL (in ICML 1998) inducing relational (Q-) trees. Around that time other work on trees, and comparisons against NNs (standard MLPs) can also be found in propositional settings (see also related seminal work by Chapman and Kaelbling in 1991 on the G-algorithm). Just to say trees have a long history in RL.

About (6): the dense nature of the paper makes it solid on one hand, but sometimes not easy to quickly see the main outcomes (especially in the experimental evaluation). I think this can be improved by making better distinctions between main and sub results. Also, some motivations, and some explanations of parts of the technical machinery in the first half of the paper could be extended to make it slightly more self-contained (although: I admit that for a technical paper like this, it is not possible entirely). I especially think that the main step from "preferences over trajectories" towards "trees of local splitting rules used to induce rewards for individual steps" can use more explanation and intuition (both technically, and conceptually).

About (7): there are many comparisons between trees and NNs for many tasks, RL or other (like supervised). The main conclusion that NNs are typically better but trees more explainable is not surprising. The paper does not provide too many more insights into the problem, and many other smalle results often come with phrases like "This could be" and "This may indeed" with some general ideas on why and how the results come about. I do want to emphasize the results in the paper are solid and systematically carried out, but I am looking for new insights on either the method/concepts as a whole (and trees vs NNs) or on the comparison of the new additions vs the original BL (but this last aspect is not part of the paper). So, it seems many experiments are done, but without too much to gain from it, is my feeling.
Also, I feel that the paper could benefit from first doing the comparison on a much simpler domain, where trees could be induce with very simple structure, to highlight aspects of the method, but also aspects of the evaluation methodology, and as a simple illustration and sanity check. With the more complex domain used, not all phenomena observed in the results can actually be fully understood/explained.


**Summary Of The Paper:**

This paper takes a published method by Bewley and Lecue (2022) named BL, which is about reward tree learning for reinforcement learning (RL) problems using preferences over trajectories as input, extends it slightly with some algorithmic tweaks and a vanilla incorporation of it inside a model-based reinforcement learning (MBRL) loop, and provides an extensive experimental evaluation with the main aim to compare against neural networks for the same task, and to conclude that these neural networks are still often the better choice but less capable of providing transparency and explainability than tree structures.

**Summary Of The Review:**

Solid paper, with a very extensive evaluation (on a single domain though, and no real comparison against the original BL algorithm) and well written, and mostly well motivated and place in the related literature. However, the novelty and significance are limited, compared to the state-of-the-art.

---

> ### Author Response · Authors · 2022-11-11
> **Response to Reviewer aaeK (Part 1)**
>
> Dear Reviewer aaeK, thanks for taking the time to post your review!
>
> **As a first port of call, we invite you to read our two-part global response, which addresses concerns raised by all three reviewers.**
>
> Below are responses to additional points made by yourself. Follow-up questions for you are highlighted in **bold**:
> - *"No real comparison against the original BL algorithm"* - This seems to be a key point to you, since you raise it three times, so it is important that we clarify that our experiments *do* evaluate the effect of the major departure from the BL algorithm: the change in splitting criterion. We find that splitting using the 0-1 loss, instead of by variance minimisation, leads to a significant gain in performance on multiple metrics. We also run an experiment highlighting the benefit of switching from model-free to model-based RL in Appendix B. Other algorithmic changes are made primarily for consistency with the wider literature rather than performance, so are not subject to evaluation.
> - *"I think it would be good to highlight the algorithmic and performance-wise aspects of these parts more explicitly in a kind of contributions"* - Given the space constraints we are working under, we propose to add this list of methodological contributions as an appendix. **Would this be acceptable for you?**
> - *"I do miss these explicit placements of the current method in the state-of-the-art"* - This is a good point; we found this a little difficult when preparing the paper. Since BL were the first to consider using trees to learn reward functions (rather than policies, value functions or dynamics models), our related work necessarily straddles multiple rather distinct fields. That said, we will attempt to add a couple of explicit references to the positioning of reward trees in the related work section.
> - *"Tree-based models in RL have a long history, and it would be good to connect to much more older work"* - Another valid point; since our related work is so diverse we had to be selective about which connected fields to acknowledge. We will seek to add a couple of references similar to the ones you give, although this may necessitate removing others to make room.
> - *"The dense nature of the paper makes it solid on one hand, but sometimes not easy to quickly see the main outcomes"* - As you know, this is a perennial tradeoff of the conference paper format, and we did our very best to find a good compromise. That said, we will review the results section with a view to bringing the main outcomes to the fore, perhaps through selective use of bold text.
> - *"I especially think that the main step from "preferences over trajectories" towards "trees of local splitting rules used to induce rewards for individual steps" can use more explanation and intuition"* - Figure 1 was included to provide such intuition; it shows how the timesteps from each trajectory that lie within each leaf end up affecting the leaf-level reward predictions. We are keen to work with you to identify the conceptual sticking point here; **could you pinpoint where in the text (+ Figure 1) your confusion begins to arise?**
>
> [Continued in Part 2...]

---

> > ### Author Response · Authors · 2022-11-11
> > **Response to Reviewer aaeK (Part 2)**
> >
> > [...Continuation from Part 1]
> >
> > - *"There are many comparisons between trees and NNs for many tasks, RL or other (like supervised). The main conclusion that NNs are typically better but trees more explainable is not surprising"* - In one sense, we do not mind this characterisation of our work as complying with a large body of evidence about the tradeoffs of interpretable models. However, with respect, we do not feel it is at all obvious from the supervised learning literature than an efficient algorithm for preference-based reward tree learning exists, and is competitive with NNs on a challenging problem of industrial relevance. The basis, function and construction method of a reward tree differs greatly from a vanilla regression tree. The merits of alternative model architectures need to be investigated anew when the context changes, and we believe we have successfully begun to do so for reward learning. Moreover, we think your takeaway that *"NNs are typically better"* belies the fact that the *margin* by which they are better seems to be much less than what one might naively expect. This is important, because it gives us permission to try using reward trees in other complex contexts, and reap the interpretability benefits they provide. Similarly, the general statement that *"trees more explainable"* ignores the nuance of context. Special opportunities for tree-based explanation present themselves when the underlying data source is preferences over trajectories. Some of these opportunities are explored for the very first time in our paper.
> > - *"I feel that the paper could benefit from first doing the comparison on a much simpler domain"* - Our intention was for the toy example in Figure 1 to provide the requisite intuition for how reward trees work. In addition, one of the domains considered by BL was a 2D problem where the partition structure could be visualised directly; referring to this may be helpful. However, we acknowledge that you're asking for the NN-tree comparison itself to be run on a simpler domain. **Could this be a major factor in changing in your view of the paper?** While we could certainly include this in a camera-ready version, we have to be candid in saying that running a full extra experiment before the rebuttal deadline would be a stretch! If this is a major point for you, we will see what can be done to address it.

---

> > > ### Comment · Reviewer_aaeK · 2022-11-15
> > > **response 2**
> > >
> > > I only realized later (also based on reading the reviews, the rebuttal, etc) that this specific setting has indeed more novel elements concerning RL+trees, in terms of the focus on reward learning, and more specifically from preference data over trajectories. Throughout RL history, policies/behaviors and value functions have been targeted almost exclusively, instead of the reward function. However, it would be good to frame this better in the related work section: 1) by highlighting the relations with inverse RL (as one of the other reviewers indicates) and 2) by raising the possibility to learn rewards from actual examples with some form of structured (tree) regression (thus, sidestepping the preference data setting). Work such as this recent paper (https://arxiv.org/abs/2010.03694) could be relevant, although I do not have references specifically to older work doing this with standard reward learning with trees (from actual reward data, thus from the agent's point of view, and not from an observer's point of view which allows for preference based evaluation of complete trajectories).
> > >
> > > My remarks and evaluation of the relative low significance/novelty of the tree vs NN results remain fairly the same, although I see your point (also in the global reply).
> > >
> > > The additional experiment is not necessary. (although I would not be against it, but it is not vital for my evaluation). A better, more practical, solution would be to try to improve the connection between text and Figure 1, I suppose.
> > >
> > > In fact, based on the global response, the constructive answers here, and re-reading parts of the paper, I still think that the paper is solid but novelty and significance are not very high, but I do feel that I have seen enough evidence to raise my evaluation from weak reject to weak accept.

---

> > > > ### Author Response · Authors · 2022-11-16
> > > > **Combined reply to "response 1" and "response 2"**
> > > >
> > > > Thanks for your continued engagement, and preparedness to raise your score. It seems your remaining concerns are primarily about (1) methodological novelty, but also about (2) the highlighting and clear description of the methodological contributions we *do* make.
> > > >
> > > > For (1), we don't expect to change your view fundamentally with the revised paper version (as the methods we use are necessarily fixed) but we are encouraged that you appreciate our perspective that novelty may not *need* to be primarily methodological. Novelty could come in the form of a new perspective, finding or scope of evaluation, all of which we believe our paper delivers.
> > > >
> > > > For (2), we hope that the paper changes we've already promised will help. In addition, in response to your comments we will now include an annotated version of Figure 1 in the appendix. This will describe the four model induction stages in greater detail (especially the trajectory-level -> transition-level credit assignment step that you found challenging).
> > > >
> > > > Additionally, we are hopeful that clarifications and additions to address (2) will in turn give you a more favourable view of (1); the changes we make to BL's method are narrow and targeted rather than structural, but this does not make them insignificant, and their implementation required significant forethought and engineering effort.
> > > >
> > > > Further to your comment about the additional experiment, we will put this on the back burner but will bear it in mind as a possible addition to a camera-ready version.

---

> > ### Comment · Reviewer_aaeK · 2022-11-15
> > **response 1**
> >
> > This is all clear, and I agree with most of it. With "no real" comparison, I guess I mean something similar as what you say here, in some sense. I guess for me it coincides with the remarks about novelty and contributions; if more clear on any of them, it helps the other. In this case, it helps if you (more clearly) define what is "extra" compared to BL in the technical machinery, and how that (empirically) helps. Here you mainly focus on the splitting (which is a main influence ofcourse). The suggestion about contributions as an appendix seems ok, but in addition I also prefer even more clarity in the introduction section (for example) to highlight the main ones.
> > About the confusion mentioned: when rereading, I notice it is about the step from trajectories to individual steps, through the "ostensibly naive" solution mentioned on page 4. It comes from BL, but it could deserve more explanation and description here, since it is a non-standard approach, and it is not obvious from this short description that it actually makes sense. Maybe here you could even illustrate with some example (although I realise it is not that simple to do, buy maybe).

---

### Official Review · Reviewer_M2ur · 2022-10-26

**Confidence:** 3
**Correctness:** 3
**Technical Novelty And Significance:** 3
**Empirical Novelty And Significance:** 3
**Recommendation:** 5

**Clarity, Quality, Novelty And Reproducibility:**

The paper has decent clarity and quality.  The novelty seems a bit limited but it's a complete pipeline from preference labels to agent training. The reproducibility should be good since they provided source code.

**Details Of Ethics Concerns:**

The experiments are conducted with an aircraft-controlling domain, where the agent maneuvers an aircraft to follow/chase/land according to a reference aircraft. Though it's not explicitly stated in the paper, it's quite likely the use case is for wars/weapons/protection of homeland/etc. For evaluating the technical aspects of the proposed method, I don't think it really must use such a domain which could cause huge ethics concerns.

**Strength And Weaknesses:**

Strength
+ A complete pipeline for tree-based reward learning from preference labels where the authors show effective learning and competitive performance against commonly used neural network approaches;
+ Detailed description of the four-stage pipeline;
+ Competitive performance is shown in evaluation against NN approaches;
+ Example reward trees are shown to understand the advantages of the tree-based reward methods;

Weaknesses
- The method is tested upon a not commonly used domain, i.e., aircraft handling environment. It's hard to understand the properties of the domain and the difficulty of learning just from the text description. The authors could use additional relatively standard benchmarks for their experiments;
- The generality of the proposed method could be shown by using more diverse domains for the experiments.


**Summary Of The Paper:**

In this paper, the authors work on preference-based RL setups, where they develop reward trees from the preference labels. They use four stages for the learning, first estimating the returns for trajectories, then, predicting the reward at the leaf level, and finally, they apply tree growth and tree pruning operations for the reward trees. Even though they observe performance hit when switching from neural network-estimated rewards to tree-based rewards, the authors argue that the tree-based methods can be competitive. The advantages of the tree-based method are shown in Figure 6 by easy interpretation of the reward meanings and semantic understanding.

**Summary Of The Review:**

I'm currently leaning towards rejection of the paper since it is showing a certain level of ethical concerns. The domains tested in the paper could be used for wars, weapons, etc, which I don't think it's the only application. The authors could demonstrate the effectiveness of their method in a less sensitive domain.

---

> ### Author Response · Authors · 2022-11-11
> **Response to Reviewer M2ur**
>
> Dear Reviewer M2ur, thanks for taking the time to post your review!
>
> **As a first port of call, we invite you to read our two-part global response, which addresses concerns raised by all three reviewers.**
>
> Below are responses to additional points made by yourself:
>
> - *"It's hard to understand the properties of the domain and the difficulty of learning just from the text description"* - Thank you for highlighting this. Although a full description of the aircraft handling domain is given in Appendix C.2, we assume you are referring to the text in the main body of the paper. We will seek to improve our wording here, and add further annotation to Figure 2.
> - *"I'm currently leaning towards rejection of the paper since it is showing a certain level of ethical concerns"* - You take issue with our application domain of aircraft handling. It seems essential that we reassure you on this matter, since your phrasing implies that you would otherwise have recommended acceptance. We kindly request that you review the conference's code of ethics (https://iclr.cc/public/CodeOfEthics), which contains no recommendation to reject any work which might plausibly have a military application. (That would preclude all mentions of aerospace, path planning, mobile robotics, computer vision, signal processing, etc.) Our application is solely concerned with the safe and humanlike control of aircraft in environments with other aircraft, which applies equally well to the uncontroversial domain of aerobatics, and with very minor alterations, civilian transportation at ground or sea. This application has absolutely no direct implication of harm. (Even if it had, the ICLR code of ethics states that *"when harm is an intentional part of the system, those responsible are obligated to ensure that the harm is ethically justified"*. Many would argue that your example of *"protection of homeland"* is precisely such an ethical justification.) Ultimately, our choice of application was made because the motivation for reward learning is clear, the core concepts of stable continuous control are widely transferable, and the resultant figures are intuitive for anybody familiar with aircraft and piloting. However, as a means of addressing your concern head-on, we will add an Ethics Statement discussing this choice of application after the main text and before the references, as suggested by the ICLR author guide.

---

### Author Response · Authors · 2022-11-08
**Reviews acknowledged; responding to each reviewer shortly**

Thanks to all reviewers for your comments! We will shortly be posting an initial response to each reviewer individually, with the aim of addressing their immediate concerns and beginning a discussion that will hopefully leave them more prepared to accept our paper. We expect to make revisions to the paper as required, once we understood how these can best be targeted to address reviewer concerns.

---

### Author Response · Authors · 2022-11-11
**Global Response: Contributions and Evaluation Domain (Part 1)**

We appreciate the perspectives of all three reviewers, and the time taken to provide constructive comments. All seem to agree that the problem of learning interpretable reward trees from human preferences is a pertinent one, our work and results are solid, and our positive conclusion about the method's performance is appropriate. In this two-part global response, we address the two most significant concerns raised by all reviewers. We complement the global response with more targeted replies to additional points raised by each reviewer.

---

Firstly, it seems we have struggled to communicate the main contribution of this work. With reference to the algorithmic updates that we make to the original proposal of Bewley & Lecue (BL), reviewers note that these changes are relatively minor, and hence our contribution is limited.

Before we present our primary response, we note that our results show that the most noteworthy of our changes - splitting based on 0-1 loss instead of variance - leads to a robust increase in both quantitative and qualitative importance. On a practical level, its efficient implementation also required a fundamental reformulation of the decision tree induction algorithm, which we do not emphasise in the paper but was actually a major aspect of our technical work. We will seek to make the technical significance of this change clearer in the wording of the paper.

Nevertheless, if this and other algorithmic updates were indeed our main contribution, we would agree with the reviewers that the overall significance was limited. **Crucially, this is not how we meant to position the paper**. As we will strive to make clearer in an updated version, our primary objective was to investigate a research question that can be phrased as follows:

- *"Can reward trees learnt from preferences perform competitively with neural networks (NNs) in a realistic and industrially-relevant domain, while remaining interpretable?"*

BL's original work focused on simple and poorly motivated domains (including standard benchmarks), used a very limited set of metrics, lacked any qualitative behaviour analysis or sensitivity study, and did not benchmark against NNs as the de facto standard. The main contribution that we wish to emphasise is thus our deep evaluation, which differs from BL's in all of these respects. We find that reward trees with around 20 leaves can achieve quantitative and qualitative performance close to that of NNs. The NN-tree gap reduces as the ground truth reward becomes more nonlinear, and remains stable or reduces further in the presence of limited or corrupted data. The extended example in Figure 6 also shows multiple ways in which reward trees, and their origins in preference data, can be analysed and understood, none of which are possible with NNs. Ultimately, we felt able to answer our research question in the affirmative.

**The central issue that reviewers should consider is whether *this* is an important contribution, and we firmly believe that it is.**

In recent years, reward learning work has adopted black box NNs as an unquestioned default. As discussed in the Introduction, we believe this to be a high-risk state of affairs, with possible real-world safety implications, because of how the lack of interpretability hampers the verification, debugging and explanation of an agent's alignment with human preferences. Our finding that tree models can be competitive is the first substantive evidence that another way is possible, and worth pursuing further.

As outlined in the last paragraph of the Introduction, this primary contribution is supplemented by a cluster of secondary ones. These include the updates to the tree induction algorithm, as well as novel aspects of our evaluation and interpretability pipelines, that we hope will be valuable to others working in the area of reward learning.

In summary, we believe our paper's contributions go beyond the technical model induction updates to say something general and important about the future of reward learning. We earnestly invite reviewers to consider this wider framing.

[Continued in Part 2...]

---

> ### Author Response · Authors · 2022-11-11
> **Global Response: Contributions and Evaluation Domain (Part 2)**
>
> [...Continuation from Part 1]
>
> The second common area of concern is our use of a single, non-standard evaluation domain. We can give multiple defences of this decision, which we hope will collectively prove satisfactory:
> - In their earlier work, BL evaluated reward tree learning in standard classic control environments from OpenAI Gym, and showed it to be effective. We refer reviewers to that paper for detailed results, and argue that replicating this work would have been an inefficient use of space in the present paper.
> - Rather than starting with a well-behaved conventional RL environment and retrofitting an artifical reward learning problem into it, we feel it is genuinely more scientifically meaningful to evaluate our methods in contexts that are plausible and industrially-relevant, where ambiguity in the correct specification of reward is a real practical concern. We refer reviewers to Appendix C.1, which gives a broader perspective on why the aircraft handling context is interesting.
> - While our three evaluation tasks are unified by the semantic context of aviation, which aids the narrative of our results and discussion, they differ extensively in the details. Points of difference include the complexity and nonlinearity of oracle reward functions, the degree of exploration required to find good solutions, and the presence or absence of termination conditions. Our evaluation thus has broader coverage than reviewers seem to imply.
> - We believe there is room in the ML literature for evaluative depth as well as breadth. We could have dedicated space in our paper to describing and motivating five or more unrelated applications, but we remain convinced that it was better to present a thorough set of quantitative, qualitative and interpretability results. We feel the outcome is a more holistic understanding of what can be achieved with reward tree learning, and a clearer answer to our research question. (To reiterate: we hold this view only in light of the fact that BL have already evaluated reward trees on multiple standard benchmarks. If a broad evaluation had not previously been conducted, we would agree that it was an essential requirement for our paper).
> - If reproducability is part of the reviewers' concerns, it need not be. Our supplementary material includes generalised, modular and optimised implementations of our model induction algorithm, which is immediately compatible with other evaluation domains, as well as of the aircraft handling environment itself. The environment implementation follows the standard OpenAI Gym schema, so should itself be compatible with researchers' existing workflows for reward learning and agent training.
> - Similarly, if reviewers are worried about a lack of comparability to existing reward learning techniques, such comparison was precisely the aim of the present work. We benchmarked reward trees against the de facto standard of NN-based reward learning throughout, replicating the NN architecture used in a well-cited recent work [1]. Our positive conclusions are thus drawn not in isolation from the wider literature, but through explicit reference to the state-of-the-art.
>
> With fear of stating the obvious, the above does not deny the utility of evaluating reward trees in other domains. Clearly this will be a beneficial line of work, to which we intend to contribute ourselves in future publications.
>
> [1] Lee, Kimin, Laura Smith, and Pieter Abbeel. "PEBBLE: Feedback-Efficient Interactive Reinforcement Learning via Relabeling Experience and Unsupervised Pre-training." In International Conference on Machine Learning. 2021.

---

> > ### Comment · Reviewer_aaeK · 2022-11-15
> > **just**
> >
> > I just want to acknowledge the clarity of this global response. I am still evaluating the other remarks, replies and plans and I can't say how/if it will change my overall opinion, but I think what the authors do here is clear and honest, and it makes sense how they want to (emphasize how to) frame the paper and its contributions.

---

> > > ### Author Response · Authors · 2022-11-15
> > > **Thanks for your acknowledgement**
> > >
> > > Dear Reviewer aaeK,
> > >
> > > Thanks for this initial comment - we'll be continually monitoring OpenReview over the remaining days until the end of the discussion period. We'll aim to respond to any follow-up queries that you have as promptly as possible!

---

### Author Response · Authors · 2022-11-11
**List of Planned Paper Changes**

Having provided our initial round of global and reviewer-specific responses, we now collate the set of changes that we have promised to make to the paper, and list them in order of their likely positioning in the document itself. We will uploaded a revised version of the paper before the end of the rebuttal/discussion phase (18th of November).

1. Make changes where necessary to emphasise that the evaluation, analysis and interpretability, rather than the model induction methodology, is the core contribution of the paper.
2. Add explicit mentions of the positioning of reward trees relative to the state-of-the-art in the related work section.
3. Add citations relating to the long history of tree-based models in RL.
4. Acknowledge the existence of differentiable tree models and justify our decision not to use them.
5. Improve the description of the NLL loss function.
6. Improve the description of the post-normalisation step during return estimation.
7. [*Hopefully informed by discussion with Reviewer aaeK*] Improve the description of the key model induction step that moves from preferences over trajectories to leaf-level reward predictions.
8. Clarify that splitting using the 0-1 loss necessitates a nontrivial reformulation of the tree growth process, which is a significant methodological contribution.
9. Clarify that the claim about model-based RL's sample complexity refers to the complexity of reward learning itself (excluding initial dynamics learning), and why this is a valid measure.
10. Improve the textual description of the aircraft handling environment in the main paper, complementing with further annotations in Figure 2.
11. Tighten up the results section with a view to bringing the main outcomes to the fore, perhaps through selective use of bold text.
12. Add an Ethics Statement discussing our choice of evaluation domain.
13. Add an appendix listing all algorithmic changes vs Bewley & Lecue's (BL's) original method, and their performance implications.
14. [*Stretch goal, dependent on response of Reviewer aaeK*] Add an appendix containing results and comparison for a much simpler and more easily-visualised domain.

---

> ### Author Response · Authors · 2022-11-16
> **Updates to Planned Changes**
>
> In light of ongoing discussions with Reviewer aaeK, we are making the following updates to our planned paper changes:
>
> - Add change 15: Include an annotated version of Figure 1 in an appendix. This will provide further detail on the four model induction stages with the aid of the toy example.
> - De-prioritise change 14 (additional experiment appendix): Reviewer aaeK considers this "not necessary". If applicable, we will revisit this potential addition when preparing a camera-ready version of the paper.

---

### Author Response · Authors · 2022-11-18
**Revised paper version uploaded; thanks again to all reviewers**

We have now uploaded a revised paper version containing changes 1-13 and 15 from the plan outlined in our earlier comments (note: change 14 was de-prioritised after feedback from Reviewer aaeK). For convenience, we temporarily highlight content that has been changed in blue text. In addition to wording revisions throughout the paper, we have produced an improved version of Figure 2, two new appendix sections, and an Ethics Statement. We have also made use of selective bold text to highlight some of the key findings in the Experiments and Results section.

We are grateful to all three reviewers for their feedback, and to Reviewer aaeK in particular for engaging constructively with us throughout the discussion period so far and taking the time to raise their evaluation. We will be continually monitoring the OpenReview page throughout the remainder of the discussion period (Nov 18 - Dec 12) and are very happy to respond to any further questions and comments that arise.

Kind regards,

Paper authors

---

### Decision · Program_Chairs · 2023-01-20

**Decision:**

Reject

**Justification For Why Not Higher Score:**

The value add of this paper is interpretability, however the interpretability of trees vs NNs is already known, and the relative performance of trees vs NNs has been investigated in other domains (e.g. in finance). Given that this paper is also targeting a very particular application domain with potential ethical implications, it does not seem to me that this paper in its current form will be a meaningful contribution to the community.

**Justification For Why Not Lower Score:**

NA

**Metareview: Summary, Strengths And Weaknesses:**

This paper proposes to replace neural network-based learnt reward models (RMs) with tree-based RMs due to the interpretability properties of the latter. The algorithm is closely based on prior work of Bewley and Lecue (2022) (BL), and the approach is evaluated on a number of tasks in the aircraft handling domain. This paper runs solid empirical evaluation of the proposed method and shows that it does work in the tested domain, resulting in interpretable RMs. However, the methodological innovation proposed in this paper is seen by the reviewers as minor (the base BL algorithm is modified only slightly), the proposed approach performs worse than NNs (although the tree-based approach is more interpretable), and the method is only evaluated on a single novel domain, which makes it hard to understand the value of the approach more broadly against the existing body of literature. Although the authors did a great job participating in the discussion period, unfortunately the reviewers remained not fully convinced that the paper merits acceptance at this point.